# ModHiFi: Identifying High Fidelity predictive components for Model Modification

**Dhruva Kashyap**
CSA, IISc
kdhruva@iisc.ac.in

**Chaitanya Murti**
HP Inc. AI Lab
chaitanya.murti@hp.com

**Pranav Nayak**
CSA, IISc
pranavk@iisc.ac.in

**Tanay Narshana** [*]
Google
tanaynarshana@google.com

**Chiranjib Bhattacharyya**
CSA, IISc
chiru@iisc.ac.in

## Abstract

Open weight models, which are ubiquitous, rarely provide access to their training data or loss function. This makes modifying such models for tasks such as pruning or unlearning, which are constrained by this unavailability, an active area of research. Existing techniques typically require gradients or ground-truth labels, rendering them infeasible in settings with limited computational resources. In this work, we investigate the fundamental question of identifying components that are critical to the model's predictive performance, without access to either gradients or the loss function, and with only distributional access such as synthetic data. We theoretically demonstrate that the global error is linearly bounded by local reconstruction errors for Lipschitz-continuous networks such as CNNs and well-trained Transformers (which, contrary to existing literature, we find exhibit Lipschitz continuity). This motivates using the locally reconstructive behavior of component subsets to quantify their global importance, via a metric that we term *Subset Fidelity*. In the uncorrelated features setting, selecting individual components based on their Subset Fidelity scores is optimal, which we utilize to propose **ModHiFi**, an algorithm for model modification that requires neither training data nor access to a loss function. **ModHiFi-P**, for structured pruning, achieves an 11% speedup over the current state of the art on ImageNet models and competitive performance on language models. **ModHiFi-U**, for classwise unlearning, achieves complete unlearning on CIFAR-10 without fine-tuning and demonstrates competitive performance on Swin Transformers.[2]

## 1 Introduction

Modern deep learning has made significant strides in a wide variety of tasks, such as classification [35, 36], image generation [22], and natural language processing [49]; moreover, *well-trained* open weight models for such tasks are easily accessible. However, significant challenges remain in their deployment, such as inference in resource-constrained settings [59, 68], inference with unbalanced or biased data [28, 29], and interpretable inference [93]. These challenges have increased interest in methods that modify the parameters of well-trained models to alter their behavior [61, 63, 65]. These methods include pruning [27], classwise unlearning [31, 34, 65], and debiasing [50, 65], among other model modifications. Moreover, recent work has studied model modification in the setting where the original training data and loss function are unavailable [53]; this is motivated by concerns

---

[*]Author primarily contributed to this work before joining Google.
[2]Our code is available at https://github.com/DhruvaKashyap/modhifi

related to privacy and security [88], and also the use of *synthetic data*, which has become critical in a variety of language modeling settings [10, 72]. Thus, we address the challenging problem of *altering well-trained models without training data or the loss function, and only with distributional access to the original training distribution in the form of synthetic data*, focusing specifically on structured pruning and classwise unlearning.

Modifying open weight models without the loss function and only synthetic data requires answering a fundamental question: *which components in a model contribute significantly to its predictive performance*[3]*?* However, most methods that identify critical components for specific modifications (e.g., pruning) cannot be applied to others (e.g., unlearning) [53], often require expensive fine-tuning, and are architecture-specific. Moreover, most methods utilize gradients to assess the impact of a component on the loss objective, which is not feasible in the absence of the loss function and the training data. While the LLM pruning literature uses calibration datasets to mitigate the problem of the absence of datasets [2, 46], the problem of achieving sparsity in vision models without original training data is hard and unsolved [27]. Moreover, the issue of performing classwise unlearning without access to the original training data has not been addressed [32, 53].

Towards enabling the modification of well-trained open weight models amidst these challenges, we make the following contributions:

(C1) **Local-to-Global with Lipschitzness**. An open question is the extent to which local model modifications impact the predictive performance of the model. In the absence of loss functions and training sets, estimating the impact of component modification by using gradients (as done in [32, 43, 46]) is infeasible. To address this, in Theorem 3.6, we show that for Lipschitz continuous networks, the reconstruction error at the final layer is at most linear in the local reconstruction errors. Moreover, contrary to the assertion that transformers are not Lipschitz continuous [60], in Corollary B.4, we show that this is not the case for *well-trained* transformers, allowing us to apply Theorem 3.6 to not just CNNs, but well-trained ViTs and LLMs as well.

(C2) **Identifying Subsets of Important Components**. Contrary to prior work, which usually infers saliencies for single components, we propose measuring the importance of sets of components to understand the cumulative effects of groups of components on a model's predictive performance. Leveraging Theorem 3.6, we propose *Subset Fidelity*, which quantifies the extent to which a subset of components can reconstruct the output after modifying their weights. However, computing optimal subsets is NP-complete, motivating us to compute Subset Fidelity scores for singleton sets. Theorem 3.9 establishes that selecting singletons with the highest subset fidelity scores is optimal when the features are uncorrelated.

(C3) **Modifying Models with ModHiFi-X.** Motivated by Theorem 3.9, we propose the **ModHiFi** algorithm, which uses the subset fidelity of singletons to modify models for pruning and classwise unlearning; the algorithm identifies important components using the singleton scores, and removes them (for classwise unlearning, **ModHiFi-U**) or retains them (for structured pruning, **ModHiFi-P**). We demonstrate that ModHiFi-P achieves state-of-the-art speedup for ImageNet models and consistently competes with current baselines for language models. For classwise unlearning, ModHiFi-U achieves complete unlearning on all CIFAR-10 classes without fine-tuning and is competitive with baselines on Swin-Transformers that require fine-tuning. When allowing for a similar fine-tuning budget as said baselines, ModHiFi-U outperforms, particularly when given access to training data. These empirical results demonstrate the practical effectiveness of Subset Fidelity.

## 2   Background, Setup, and Related Work

In this section, we review the background relevant to our study, establish the notation, and formalize the model modification problem. We also unify Convolutional Networks (CNNs) and Transformers under a single abstraction that underpins our theoretical results in Section 3.

---

[3]We measure predictive performance using accuracy for classification tasks, and perplexity and other measures for language modeling tasks.

## 2.1 Background and Notation

**Notation**  Let $[p] = \{1, \ldots, p\}$ for $p \in \mathbb{N}$. We denote vectors by $\boldsymbol{v} \in \mathbb{R}^n$ with entries $v_i$, and matrices by $\boldsymbol{B} \in \mathbb{R}^{n \times m}$ with rows $\boldsymbol{b}_i^\top$ and columns $\boldsymbol{B}_{:,j}$. The vectors $\mathbf{1}_d$ and $\mathbf{0}_d$ denote the all-ones and all-zeros vectors in $\mathbb{R}^d$, respectively. We use $\|\boldsymbol{v}\|_2$ for the Euclidean norm. For matrices $\boldsymbol{C}, \boldsymbol{D}$, the inner product is $\langle \boldsymbol{C}, \boldsymbol{D} \rangle = \mathrm{Tr}(\boldsymbol{C}^\top \boldsymbol{D})$, the Frobenius norm is $\|\boldsymbol{C}\| = \sqrt{\langle \boldsymbol{C}, \boldsymbol{C} \rangle}$, and the spectral norm $\|\boldsymbol{C}\|_2$ is the largest singular value. For index sets $A \subseteq [n]$ and $B \subseteq [m]$, $\boldsymbol{C}[A, B]$ denotes the submatrix of $\boldsymbol{C}$ defined by these indices. Expectations of a random variable X are written $\mathbb{E}_X[\cdot]$, omitting the subscript when clear from context.

**2D Convolution**  Consider the $l$-th layer of a convolutional network. It transforms an input (the output of preceding layers) $\Phi^l(\boldsymbol{X}) \in \mathbb{R}^{c_{\mathrm{in}}^l \times h^{l-1} \times w^{l-1}}$ into output $\mathbf{Y}^l(\boldsymbol{X}) \in \mathbb{R}^{c_{\mathrm{out}}^l \times h^l \times w^l}$. The layer is parameterized by a weight tensor $\mathbf{W}^l \in \mathbb{R}^{c_{\mathrm{out}}^l \times c_{\mathrm{in}}^l \times k^l \times k^l}$. Each output channel $c \in [c_{\mathrm{out}}^l]$ is computed as the sum of convolved input channels:

$$\boldsymbol{Y}_c^l(\boldsymbol{X}) = \sum_{i=1}^{c_{\mathrm{in}}^l} \Phi_i^l(\boldsymbol{X}) \star \boldsymbol{W}_{ci}^l = \sum_{i=1}^{c_{\mathrm{in}}^l} \boldsymbol{A}_{ci}^l(\boldsymbol{X}), \tag{CONV}$$

where $\star$ denotes the standard 2D convolution. We define $\boldsymbol{A}_{ci}^l(\boldsymbol{X}) := \Phi_i^l(\boldsymbol{X}) \star \boldsymbol{W}_{ci}^l \ (\in \mathbb{R}^{h^l \times w^l})$ as the *input contribution* from channel $i$ to output channel $c$. For notational simplicity, we omit explicit bias terms and stride/padding specifications, as our analysis generalizes to these standard configurations without loss of generality.

**Transformers**  Transformer blocks consist of Multi-Head Attention (MHA) and Feed-Forward Networks (FFN), with pre-normalization (LayerNorm or RMSNorm) [2, 3]. Our analysis focuses on the FFN; we leave attention-specific analysis for future work. Let the input to the $l$-th layer be $\phi^l(\boldsymbol{X}) \in \mathbb{R}^{T \times d}$, where $T$ is sequence length and $d$ is model dimension. The FFN comprises two linear transformations, $W_U^l \in \mathbb{R}^{d \times d_{\mathrm{ff}}}$ and $W_D^l \in \mathbb{R}^{d_{\mathrm{ff}} \times d}$, and an elementwise nonlinearity $\sigma(\cdot)$ and it's output, $\mathrm{FFN}^l(\phi^l(\boldsymbol{X})) = \sigma(\phi^l(\boldsymbol{X})W_U^l) W_D^l$. Defining the intermediate activation $\Phi^l(\boldsymbol{X}) := \sigma(\phi^l(\boldsymbol{X})W_U^l) \in \mathbb{R}^{T \times d_{\mathrm{ff}}}$, the contribution from intermediate neuron $i \in [d_{\mathrm{ff}}]$ to output coordinate $c \in [d]$ is:

$$\boldsymbol{A}_{ci}^l(\boldsymbol{X}) := \Phi_{:,i}^l(\boldsymbol{X}) \, W_{D,ci}^l. \tag{LIN}$$

**Unified Notation**  To unify these architectures, we define a common abstraction used in our theoretical results. Let $\mathrm{N}_\theta = f^L \circ \cdots \circ f^1$ be a network composed of $L$ layers. Each layer $l$ maps an input $\Phi^l(\boldsymbol{X})$ to an output $\mathbf{Y}^l(\boldsymbol{X})$. Crucially, for both CNNs and Transformers, the output channel $c$ can be decomposed as a sum of atomic input contributions: $\boldsymbol{Y}_c^l(\boldsymbol{X}) = \sum_{i=1}^{c_{\mathrm{in}}^l} \boldsymbol{A}_{ci}^l(\boldsymbol{X})$, where $\boldsymbol{A}_{ci}^l$ represents either the spatial convolution (Equation (CONV)) or the token-wise linear projection (Equation (LIN)). This decomposition is central to our analysis of component importance. This additive structure allows us to analyze component fidelity in an architecture-agnostic manner.

## 2.2 Modifying Open Weight Models without Training Data or the Loss Function via Distributional Access

We formally define *model modification* as the process of selectively altering parameters of a pre-trained model, without retraining from scratch, to satisfy constraints such as efficiency, privacy, or safety [27, 32, 63, 65]. This includes tasks including structured pruning, unlearning [37], debiasing [31], continual or life-long learning [21, 61]. A major impediment in real-world modification is the unavailability of the original training data and loss function [27, 53]. To address this, we operate under the constraint of *distributional access*, specifically utilizing *synthetic data* [10, 72], to proxy the underlying data distribution without requiring the original corpus.

These considerations motivate the central question addressed in this work: *Can we effectively modify trained models, for tasks such as structured pruning or unlearning, using only distributional access provided through synthetic data?*

**Formulating Model Modification**    Let $\theta^\star \in \mathbb{R}^D$ be the parameters of a well-trained model. We seek a modification mask $\mathsf{m}^\star \in \mathcal{M}$ (where $\mathcal{M}$ defines permissible modifications, e.g., binary masks for pruning) to produce modified parameters $\theta^E = \theta^\star \odot \mathsf{m}^\star$. Given data distributions $\{\mathcal{D}_i\}_{i=1}^K$ and weights $\boldsymbol{\alpha} \in \mathbb{R}^K$, the optimal modification mask is defined as:

$$\mathsf{m}^\star = \arg\min_{\mathsf{m} \in \mathcal{M}} \sum_i \alpha_i \, \mathbb{E}_{X \sim \mathcal{D}_i} \left[ \mathcal{L}(N_{\theta^\star \odot \mathsf{m}}(X)) \right]. \tag{MODIFY}$$

We instantiate this framework for two distinct tasks:

**Structured Pruning** The goal is to maximize performance subject to sparsity. Let the parameters be partitioned into $G$ disjoint structured groups $\{\mathcal{G}_g\}_{g=1}^G$ (e.g., filters, channels, rows, or columns), with $\theta^\star = (\theta^\star_{\mathcal{G}_1}, \ldots, \theta^\star_{\mathcal{G}_G})$. The admissible set enforces a sparsity budget $B$, $\mathcal{M}_{\mathrm{SP}} := \{\mathsf{m} \in \mathbb{R}^D \mid \exists \mathsf{z} \in \{0,1\}^G, \ \boldsymbol{\delta} \in \mathbb{R}^D \text{ s.t. } \mathsf{m}_j = \mathsf{z}_g \delta_j \ \forall j \in \mathcal{G}_g, \ \sum_{g=1}^G \mathsf{z}_g \leq B\}$. Structured pruning is recovered from (MODIFY) by setting $K = 1$, $\mathcal{D}_1 = \mathcal{D}$ (original task distribution), $\alpha_1 = 1$, and $\mathcal{M} = \mathcal{M}_{\mathrm{SP}}$, yielding

$$\mathsf{m}^\star = \arg\min_{\mathsf{m} \in \mathcal{M}_{SP}} \mathbb{E}_{\mathcal{D}} \left[ \mathcal{L}(N_{\theta^\star \odot \mathsf{m}}(X)) \right]. \tag{STRUCT-PRUNE}$$

**Classwise Unlearning** The goal is to degrade performance on a *forget* distribution $\mathcal{D}_f$ while preserving performance on a *retain* distribution $\mathcal{D}_r$. We impose no additional structural constraints on the modification and set $\mathcal{M}_U = \mathbb{R}^D$. Classwise unlearning is obtained from (MODIFY) by setting $K = 2$, $(\mathcal{D}_1, \mathcal{D}_2) = (\mathcal{D}_r, \mathcal{D}_f)$, $(\alpha_1, \alpha_2) = (1, -1)$, and $\mathcal{M} = \mathcal{M}_U$, yielding

$$\mathsf{m}^\star = \arg\min_{\mathsf{m} \in \mathcal{M}_U} \mathbb{E}_{X \sim \mathcal{D}_r}[\mathcal{L}(N_{\theta^\star \odot \mathsf{m}}(X))] - \mathbb{E}_{X \sim \mathcal{D}_f}[\mathcal{L}(N_{\theta^\star \odot \mathsf{m}}(X))]. \tag{UNLEARN}$$

Our core challenge is to solve (MODIFY) using only synthetic samples, *without access to ground-truth labels or the original loss*.

## 2.3    Related Work

We briefly situate our work within the literature on vision and language model modification. A comprehensive survey is provided in Appendix A.

**Vision Model Modification** While structured pruning is well-established for CNNs and ViTs [14, 27, 89, 91], and classwise unlearning has seen recent progress [12, 32], these tasks are typically treated in isolation. Crucially, prior methods for jointly addressing these problems rely heavily on access to labeled data [53]. Our work presents the first unified framework for both pruning and unlearning, which operates effectively using only unlabeled synthetic data.

**LLM Modification** Efficiency in LLMs is primarily addressed via structured pruning [46, 47] or sparsification [2]. However, these methods are often architecture-specific and do not extend to unlearning. By validating our method on both LLMs and vision models, we demonstrate a generalized approach to model modification that bridges the gap between these distinct domains.

# 3    Which Components Are Important for Modifying Well-Trained Models?

We now address the problem of identifying model components critical to predictive performance. We introduce *Subset Fidelity*, a metric that quantifies the local reconstructive capacity of component groups and *High-Fidelity (HiFi)* components. We show theoretically that maximizing local fidelity minimizes a linear upper bound on the global predictive error.

## 3.1    High-Fidelity Components and the Subset Fidelity Score

Our objective is to estimate the impact of removing a subset of input contributions on the model's output, after optimally compensating for this removal. Directly quantifying this effect is difficult, so we introduce the *Subset Fidelity*, a measure of how well a subset of components can locally approximate the layer's output.

**Definition 3.1** (Subset Fidelity)**.** The *fidelity* of a subset of components $C \subseteq [c_{in}^l]$ in layer $l$ for output channel $c$ is defined as

$$\mathrm{FS}_c^l(C) := \max_{\boldsymbol{\delta}_c^l \in \mathbb{R}^{c_{in}^l}} \left( 1 - \frac{\mathbb{E}\left[\|\boldsymbol{Y}_c^l(X) - \sum_{i \in C} \delta_{ci}^l \boldsymbol{A}_{ci}^l(X)\|^2\right]}{\mathbb{E}\left[\|\boldsymbol{Y}_c^l(X)\|^2\right]} \right), \tag{1}$$

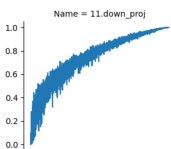 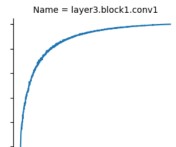 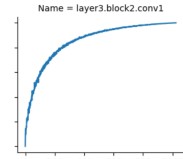 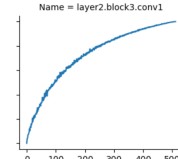

(a) OPT-125M (WikiText)    (b) ResNet50 (CIFAR-10)    (c) ResNet50 (CIFAR-100)    (d) ResNet50 (ImageNet)

Figure 1: Monte Carlo estimation of Equation (κ-MFS) across selected layers of various models. The x-axis indicates subset size $k$, and the y-axis the maximum fidelity found across random samples.

where $\boldsymbol{\delta}_c^l$ is the *compensation term*.

The following properties (proved in Appendix B.2) justify its use as an importance measure.

**Lemma 3.2** (Properties of Subset Fidelity). *For any subset $C \subseteq [c_{in}^l]$ in layer $l$, **(Boundedness)** $0 \leq \mathrm{FS}_c^l(C) \leq 1$ and **(Monotonicity)** If $D \subseteq C$, then $\mathrm{FS}_c^l(D) \leq \mathrm{FS}_c^l(C)$.*

A larger Subset Fidelity indicates that the subset more effectively reconstructs the output, thereby reducing the error of approximating the sum of components with components from a subset. Lemma 3.2 implies two key insights: (1) Fidelity serves as a principled measure of component importance, and (2) Monotonicity suggests that greedy selection strategies may be effective.

*Remark* 3.3. Equation (1) is a generalizes the formulation of El Halabi et al. [11]. In this work, we focus only on the case where the subset fidelities are measured with the expected squared difference. We leave to future work an exploration of other possible measures of distributional similarity.

To capture the tradeoff between the size of a subset and its fidelity, we define HIFI Sets.

**Definition 3.4** ($(k, \eta)$-HIFI Set). Given a target subset size $k$ and a fidelity threshold $\eta \in (0, 1)$, the $(k, \eta)$-HIFI Set $S_c^{k,\eta}$ for output channel $c$ is any subset in $[c_{in}^l]$ satisfying

$$\mathrm{FS}_c^l(S_c^{k,\eta}) \geq \eta, \quad |S_c^{k,\eta}| \leq k. \tag{HIFI}$$

Thus, attributing predictive performance to components reduces to finding the HIFI set for a given $(k, \eta)$. We can reduce the identification of HIFI sets to solving an optimization problem, the solution of which yields the *Maximum Fidelity Subset*, which contains the components that best recover the layer's output.

**Definition 3.5** ($k$-Maximum Fidelity Subset). Given a target subset size $k$ for layer $l$, the *Maximum Fidelity Subset* $S_c^{l\star}$ for channel $c$ is defined as

$$S_c^{l\star} = \underset{S \subseteq [c_{in}^l], |S|=k}{\arg\max} \mathrm{FS}_c^l(S). \tag{κ-MFS}$$

A simple algorithm for identifying a $(k, \eta)$-HIFI set is to solve Equation (κ-MFS) for the given $k$ and check whether its fidelity exceeds $\eta$. If it does not, no such $(k, \eta)$-HIFI set exists. Before proceeding to our theoretical analysis, we empirically verify whether small HiFi sets actually exist in standard models. Our experiments in Section 5.2 empirically establish the existence of a small subset of components that can achieve high fidelity. Moreover, in Section 5.3, we validate the effectiveness of HiFi components with the model's predictive performance. Figure 1 indicates a sample of the results indicating that fewer than 20% of components can achieve high fidelity ($\geq 0.8$).

## 3.2 Local Distributional Measures of Component Importance

Finding HIFI subsets corresponds to finding subsets that minimize the $l_2$ reconstruction error while accounting for weight compensation. Additionally, it enables the derivation of a closed-form expression for weight compensation, allowing for accuracy recovery without requiring fine-tuning.

**Bounding Global Error via Local Modification**    We now show that the influence of a component on its immediate layer output provides a tractable proxy for its overall effect on model predictions. The *global error* is the expectation of the squared difference in the predictions of a network under a modification.

**Theorem 3.6** (Local to Global). *Consider a network $N_\theta$ as defined in Section 2.1. Let $M^l$ be a mask modifying parameters at layer $l$, and let $\boldsymbol{m}_c^l$ be the mask vector for output channel $c$. Assume there exist scalars $r^\ell > 0$ for all layers $\ell > l$ such that $\|\boldsymbol{\Phi}_c^\ell(X)\|_F \geq r^\ell$ almost surely. Then,*

$$\mathbb{E}\left[\|N_\theta(X) - N_{\theta \odot M^l}(X)\|^2\right] \leq \mathcal{O}\left(\sum_{c=1}^{c_{out}^l} \mathbb{E}\left[\left\|\boldsymbol{Y}_c^l(X) - \sum_{i \in C} m_{ci}^l \boldsymbol{A}_{ci}^l(X)\right\|^2\right]\right) \qquad (2)$$

*Sketch.* The proof relies on the propagation of error through Lipschitz-continuous layers. See Appendix B.1. □

Theorem 3.6 upper-bounds the global error, given by the left-hand side, by a linear function of the local reconstruction errors for each channel in layer $l$. This implies that global error grows at most linearly with local error, making local fidelity a practical, architecture-agnostic proxy for component influence. The theorem requires that the networks discussed in this work are Lipschitz continuous under suitable conditions. While CNNs are known to be Lipschitz continuous [90], transformers are not [60]. In Corollary B.4, we show that this is not the case for *well-trained* transformers.

*Remark* 3.7. The leading constant in the order notation quantifies the amplification of local errors through subsequent layers and activations, and is independent of the data distribution, depending only on the model's architecture. Empirical estimates of the constant reported in Appendix C.2 demonstrate the practicality of these constants.

**Subset Fidelity for Individual Components** Next, we show that both the compensation term and the singleton fidelity scores admit closed-form expressions, thus motivating their use in this work. A derivation is provided in Appendix B.3.

**Proposition 3.8** (Compensation and Singleton Fidelity). *For the $l_2$ reconstruction error, the optimal compensation term $\delta_c^\star$, which is the value at which the fidelity score is computed according to Equation (1) for a subset $C$, is given by,*

$$\delta_{ci}^{l\star}(C) = \begin{cases} 1 + ((\boldsymbol{Q}_c^l[C,C])^{-1})_i^\top \boldsymbol{Q}_c^l[C,\overline{C}]\mathbf{1}_{n-k} & \text{if } i \in C \\ 0 & \text{if } i \notin C \end{cases} \qquad \text{(FS)}$$

*where $\boldsymbol{Q}_c^l \in \mathbb{R}^{c_{in}^l \times c_{in}^l}$ is the* component similarity matrix (CSM) *for channel $c$, with entries $(\boldsymbol{Q}_c^l)_{ij} = \mathbb{E}[\langle \boldsymbol{A}_{ci}^l(X), \boldsymbol{A}_{cj}^l(X)\rangle]$. The singleton fidelity scores are:*

$$s_{ci}^l = \mathrm{FS}_c^l(\{i\}) = 1 - \frac{\mathbb{E}[\|\boldsymbol{Y}_c^l(X) - \alpha_{ci}^l \boldsymbol{A}_{ci}^l(X)\|^2]}{\mathbb{E}[\|\boldsymbol{Y}_c^l(X)\|^2]}, \qquad \alpha_{ci}^l = \frac{\mathbb{E}[\langle \boldsymbol{Y}_c^l(X), \boldsymbol{A}_{ci}^l(X)\rangle]}{\mathbb{E}[\|\boldsymbol{A}_{ci}^l(X)\|^2]}. \qquad (3)$$

Note that solving Equation (K-MFS) exactly is still equivalent to a constrained binary quadratic optimization problem, known to be NP-hard [1]. Viewing $\boldsymbol{Q}^c$ as the adjacency matrix of a weighted graph, maximizing Equation (K-MFS) corresponds to identifying a clique of size $k$, the decision version of the MAXIMUM CLIQUE problem. Intuitively, such cliques correspond to groups of components whose joint removal maximally increases the reconstruction error.

**Computing the k-MFS** Since fidelity is monotonic, a natural heuristic selects the $k$ components with the highest singleton fidelities $s_{ci}^l$; we call this strategy: NAIVE. To compute the set of highest fidelity, the k-MFS, we identify conditions under which the NAIVE selection strategy is optimal.

**Theorem 3.9.** *Consider output channel $c$ in the $l^{th}$ layer of a network described in Section 2.1. Let the $s_{ci}^l$ be defined according to Equation (3) and $S_c^{l\star}$ be defined according to Definition 3.5. Let $\hat{S}_c^l = \{i \mid s_{ci}^l \geq s_{(k)}\}$ where $s_{(k)}$ is the $k^{th}$ largest value of $\boldsymbol{s}_c^l$. Assuming that there are no ties, $|\hat{S}_c^l| = k$. If $\mathbb{E}[\langle \boldsymbol{A}_{ci}^l(X), \boldsymbol{A}_{cj}^l(X)\rangle] = 0 \ \forall i \neq j$, then $\hat{S}_c^l = S_c^{l\star}$.*

*Sketch.* Under the assumptions, the objective simplifies from quadratic to linear. See Appendix B.4. □

*Remark* 3.10. Theorem 3.9 connects a statistical property of the representations to the efficient discovery of HiFi components. It states that when the input contributions are pairwise uncorrelated, the optimal subset is the set of components with the highest fidelity score.

Although the assumption of uncorrelated features rarely holds exactly in practice, it offers a sound theoretical justification for NAIVE HIFI selection. We demonstrate the practical effectiveness of NAIVE HIFI selection through our experiments in Section 5.

## 4 Modifying Model Behavior using HiFi Sets

We now propose MODHIFI, a unified algorithmic framework for model modification using only distributional access. We apply this framework to two distinct tasks: structured pruning (MODHIFI-P) and classwise unlearning (MODHIFI-U). The central idea is to identify high-fidelity (HIFI) components and then modify them in a targeted manner using a unified algorithmic procedure, as shown in Algorithm 1. The two tasks operate as duals: pruning *retains* the high-fidelity components necessary for general performance, while unlearning *removes* the high-fidelity components most discriminative for a specific target class. Additional details, including complexity and implementation specifics, are provided in Appendix D.

**Structured Pruning**  To address Equation (STRUCT-PRUNE), where the objective is to remove entire input channels (or features) that contribute minimally to the model's predictive performance. In convolutional architectures, we identify and remove input channels across all layers that do not appear in the HIFI sets of any output channel of the residual-coupled layers. For CNNs, pruning is applied to the input channels of convolutional layers. For LLMs, we target the input features of the MLP down-projection matrices ($W^D$). After pruning, we compute the optimal compensation term $\delta^\star$ (derived in Proposition 3.8) using the remaining weights. This step restores the fidelity of the layer output without requiring gradient-based fine-tuning.

**Class Unlearning**  The goal of Equation (UNLEARN) is to erase the influence of a specific *forget class*. To perform unlearning, we first compute HIFI sets using only samples from the class we wish to forget. The components in these sets are then zeroed out, effectively erasing the influence of that class. This causes the model's predictive performance on the forgotten class to degrade, without significantly impacting the performance of other classes.

**Fidelity Estimation**  For vision models, the singleton fidelity score $\text{FS}_c^l(\cdot)$ can be estimated efficiently using distributional access to the input data, i.e., synthetic samples. In practice, for vision models, we estimate the scalar coefficients $\alpha_{ci}^l$ directly via batched forward passes on synthetic samples. A large $\alpha_{ci}^l$ indicates a high-fidelity component. For LLMs, we develop a tractable Cholesky-based heuristic to estimate the score, providing details in Appendix D.2.

---

**Algorithm 1** ModHiFi-X

**Require:** Model parameters $\theta$, layer $l$, $k$ components, threshold $\eta$, data $\mathcal{D}$
**Ensure:** Modified parameters $\theta^E$
1: **Estimate Fidelity:** Compute singleton scores $\mathbf{s}^l$ on $\mathcal{D}$ via Equation (3).
2: **Select HiFi Set:** $H_l \leftarrow$ Top-$k$ indices of $\mathbf{s}^l$.
3: **if** $X =$ Prune **then**
4:     **for** $i \in [c_{in}^l] \setminus \{i \mid (c, i) \in H_l\}$ **do**
5:         $\boldsymbol{W}_{c,i}^l \leftarrow \boldsymbol{0} \quad \forall c \in [c_{out}^l]$
6:     Apply compensation $\delta^\star$ to remaining weights.
7: **else if** $X =$ Unlearn **then**
8:     **for** $(c, i) \in H_l$ **do**
9:         $\boldsymbol{W}_{c,i}^l \leftarrow \boldsymbol{0}$
10: **return** $\hat{\theta}$

---

## 5 Experiments

We empirically validate our framework by addressing four central questions:

(Q1) **Existence of HiFi components**. Do a small subset of components exist that can achieve high fidelity?

(Q2) **Effectiveness of HIFI components**. Do HIFI components accurately represent those components important for the predictive performance?

(Q3) **Effectiveness of using HIFI components for pruning using ModHiFi-P**. Does ModHiFi-P result in better accuracy-sparsity tradeoff compared to structured pruning algorithms for vision tasks and language modeling tasks?

(Q4) **Effectiveness of using HIFI components for machine unlearning using ModHiFi-U**. Is it possible to perform machine unlearning, as posed by Jia et al. [32], without finetuning? If so, how does ModHiFi-U compare to their method?

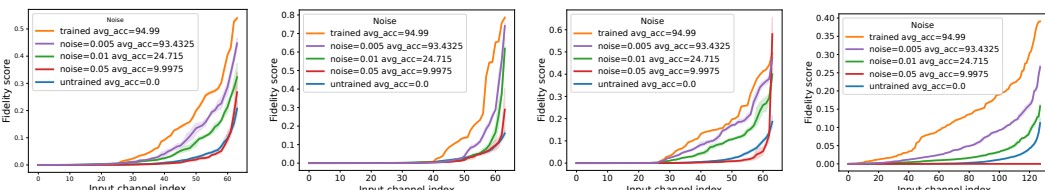

Figure 2: Fidelity score of selected layers of a ResNet-50 model on CIFAR10 and the effect of noise on the fidelity score.

## 5.1 Details of the experimental setup

**Models, Datasets, and Evaluation** We conduct experiments on ResNet-50/101 [25], VGG19 [69], Swin-Transformer [44] and Llama-2-7B [76], benchmarking against relevant experiments from related literature [2, 46]. For vision tasks, we measure the classification accuracy, and for NLP tasks, we use EleutherAI's `lm-eval-harness` [19].

**Distributional Access** For CIFAR10/100 [35], we use synthetically generated images as detailed in Appendix C.3. We use Alpaca [74] (a synthetic dataset) and WikiText-2 [48] as calibration data for NLP tasks following related literature [2, 46]. We provide **ablations** to measure the impact of synthetic data quality in Appendix C.3.3.

**Compute platform and implementation details** We discuss the compute platform, implementation details, and hyperparameters used for our experiments in Appendix C.6.

## 5.2 Existence of HIFI components: Exploring (Q1)

To empirically assess whether small subsets can achieve high fidelity, we estimate $S_c^\star$ by sampling random subsets of size $k$ across different architectures and selecting the subset with the highest fidelity. Detailed results are presented in Appendix C.1.

**Observation 1.** *Across all evaluated models, each layer typically contains a small subset of input channels (fewer than 20%) that achieves high subset fidelity ($\geq 0.8$).*

This empirical observation suggests that in trained models, only a small subset of components in each layer is responsible for the model's prediction. This observation aligns with the success of structured pruning algorithms in constructing small subnetworks with high statistical performance.

## 5.3 Effectiveness of HIFI components: Exploring (Q2)

To answer (Q2), and verify whether HIFI components are the components that matter for the final predictive performance, we measure the effects of the fidelity of a component getting destroyed by noising. For a ResNet-50 on CIFAR-10, when 20% of the HIFI components are perturbed with a zero mean Gaussian noise with standard deviation of 0.01, the accuracy of the model drops by around 12%. In contrast, perturbing 80% of the non-HIFI components identically results in an accuracy drop of only 1%. At 50% of components with a noise of standard deviation 0.02, the accuracy drops by 85% when HIFI components are noised compared to only around 1.4% when non-HIFI components are noised. In Appendix C.1.3, we make similar observations across various models and tasks. In Appendix C.1.2, we additionally performed experiments where we compare the *removal* of HIFI, non HIFI, and random sets of the same size and make similar observations.

## 5.4 Structured Pruning Experiments: (Q3)

### 5.4.1 Vision Models

**Baselines** We compare against the state-of-the-art structured pruning algorithms specialized for pruning vision models [8, 45, 56, 79], and present additional results on other architectures and datasets in Appendix C.4 where we make similar observations. Following [16], we update the batch norm statistics using the data from distributional access.

Table 1: Comparison of pruning methods on ResNet50 evaluated on ImageNet.

| Algorithm | Accuracy | FLOP Reduction | Param Reduction | CPU Speedup | GPU Speedup |
|---|---|---|---|---|---|
| Unpruned | 76.1 | 1x | 1x | 1x | 1x |
| GReg-2 [79] | 73.9 | 3.02x | 2.31x | 1.36x | 1.53x |
| OTO [8] | 74.7 | 2.86x | 2.81x | 1.25x | 1.45x |
| DepGRAPH [13] | 75.83 | 2.07x | - | - | - |
| ThiNet [45] | 71.6 | 3.46x | 2.95x | 1.38x | 1.50x |
| DFPC (30) [56] | 75.9 | 1.98x | 1.84x | 1.42x | 1.53x |
| DFPC (54) [56] | 73.80 | 3.46x | 2.65x | 2.37x | **2.38x** |
| **Ours** | **76.70** | 2.17x | 1.47x | 1.69x | 1.70x |
| **Ours** | 73.82 | **3.66x** | **3.05x** | **2.42x** | **2.38x** |

Table 2: Comparison of pruning methods on ResNet50 with CIFAR10 *(ST: Synthetic Tuning)*.

| Algorithm | Accuracy | FLOP Reduction | Param Reduction |
|---|---|---|---|
| Unpruned | 94.99 | 1x | 1x |
| DFPC [56] | 90.25 | 1.46x | 2.07x |
| $L_2$ [39] | 15.91 | **4.07x** | 4.71x |
| $L_2$ w/ ST [39] | 90.12 | **4.07x** | 4.71x |
| **Ours** | **91.02** | **4.07x** | **5.36x** |

**Observations** We find that our method yields a better accuracy-vs-sparsity tradeoff compared to other algorithms across various datasets. We also *train* a model obtained with $L_2$ norm-based structured pruning *using the synthetic set* based on CIFAR10 for comparison. In Table 2, we observe that for the *same FLOP sparsity*, our method obtains higher accuracy than the model finetuned on synthetic samples, indicating that our method can *outperform finetuning in some cases* using synthetic samples for the same sparsity. For the ImageNet dataset, we compare our approach against various state-of-the-art structured pruning algorithms for networks with complex interconnections, including those trained on the ImageNet training set. In Table 1, we observe that for models of similar accuracy, our algorithm obtains the best accuracy-speedup tradeoff with fewer epochs of finetuning. Details of pre-trained networks and post-training are given in Appendix C.7.2. Our study of the effect of the quality of synthetic samples on our algorithm in Appendix C.3.3 indicates that the sparsity-accuracy tradeoff of our algorithm degrades with lower quality samples, but it does not degrade as much as $L_2$ pruning + finetuning on synthetic samples.

### 5.4.2 Large Language Models

**Baselines** We evaluate ModHiFi on Llama-2-7B, comparing it against state-of-the-art algorithms for structured pruning [2, 47]. The use of calibration datasets to compute statistics aligns with our framing of distributional access to data, as LLMs do not make their training data openly accessible. Unless otherwise specified, the algorithms use WikiText-2 for calibration, with 128 samples of length 1024 [4]. None of the algorithms performs post-pruning recovery finetuning. Additional details about our choice of baselines can be found in Appendix C.4.3.

**Evaluation** We also measure the performance of the model via its zero-shot accuracy on a suite of standard NLP tasks [5, 9, 62, 92] and WikiText perplexity. In Table 3, we observe that our method is competitive, with consistently high average and task-specific performance, and outperforms at moderate sparsity levels. We find that the quality of the calibration set plays a crucial role, with the performance of ModHiFi-P-Alpaca outperforming that of ModHiFi-P-WikiText. This indicates that retaining only HIFI components provides a **model-agnostic** approach to structured pruning, with its application to LLMs requiring no modifications beyond its application to vision models.

### 5.5 Class Unlearning Experiments: (Q4)

**Baselines and Metrics** We report the forget and retain accuracy averaged across 10 classes of the CIFAR10 dataset on ResNet-50 and Swin-T models. We benchmark against Gradient Ascent and Jia et al. [32], which are both retraining-based techniques for Unlearning.

---

[4]ShortGPT's calibration data is not publicly available.

Table 3: Comparison of pruning methods on Llama-2-7B, measured with PPL and task accuracy

| Sparsity | Algorithm | WikiText PPL ↓ | ARC-e ↑ | ARC-c ↑ | PIQA ↑ | WinoG. ↑ | HellaS. ↑ | Average |
|---|---|---|---|---|---|---|---|---|
| 0% | Dense | 5.12 | 74.58 | 46.25 | 79.11 | 69.06 | 75.99 | 69.00 |
| 10% | SliceGPT [2] | 6.46 | 56.14 | 35.33 | 69.53 | 64.80 | 59.02 | 59.96 |
| | ModHiFi-P-WikiText (**ours**) | **5.97** | 68.1 | 41.89 | 75.89 | 65.43 | 69.92 | 64.23 |
| | ModHiFi–P-Alpaca (**ours**) | 6.36 | **71.42** | **42.06** | **76.44** | **68.19** | **71.67** | **65.96** |
| 20% | ShortGPT [47] | 14.32 | 58.33 | 38.05 | 72.58 | **65.51** | **65.27** | 59.95 |
| | SliceGPT [2] | 8.13 | 50.08 | 31.14 | 64.85 | 62.04 | 48.84 | 51.39 |
| | ModHiFi-P-WikiText (**ours**) | **7.91** | 60.1 | 34.89 | 70.62 | 61.48 | 58.7 | 57.16 |
| | ModHiFi-P-Alpaca (**ours**) | 9.38 | **64.73** | **38.22** | **72.79** | 64.64 | 62.7 | **60.62** |
| 30% | ShortGPT [47] | 33.21 | 48.65 | **32.85** | 64.31 | **64.33** | **56.13** | **53.25** |
| | SliceGPT [2] | **10.96** | 44.19 | 27.47 | 58.71 | 57.46 | 41.27 | 45.82 |
| | ModHiFi-P-WikiText (**ours**) | 11.53 | 48.98 | 28.07 | 64.03 | 55.88 | 46.19 | 48.63 |
| | ModHiFi-P-Alpaca (**ours**) | 14.78 | **53.15** | 32.5 | **66.59** | 59.35 | 50.61 | 52.44 |

Table 4: Comparison of class unlearning methods on CIFAR10.

| Model | Algorithm | Forget Acc. | Remain Acc. | Time (sec) |
|---|---|---|---|---|
| ResNet-50 | Base | 94.99 | 94.99 | - |
| | Gradient Ascent | 6.59 | 93.44 | 30 |
| | Jia et al. [32] | 3.54 | **94.14** | 363 |
| | **Ours** | **0.2** | 92.98 | **10** |
| Swin-T [44] | Base | 92.31 | 92.31 | - |
| | Jia et al. [32] | **1.20** | **90.69** | 235 |
| | **Ours** | 8.83 | 73.57 | **2** |

**Unlearning Results** We report the results of our algorithm in Table 4. To answer (Q4), we observe that it is possible to perform unlearning **without finetuning** in a general editing framework $10\times$ faster than our baseline. In Appendix C.5, we compare results with finetuning using synthetic and training data. We note that the results for Swin-Transformer without finetuning fail to achieve the state of the art. However, as reported in Appendix C.5, we observe a drastic improvement with only three epochs of finetuning on synthetic samples. After 10 epochs of finetuning with our algorithm, we find that our forget accuracy is superior to that of [32] (who use full training) when using synthetic samples. Both forget and remain accuracy are superior when using training samples. Experiments with VGG-19 are present in Appendix C.5 where we make similar observations.

## 6 Discussion and Conclusion

We have addressed the challenge of modifying well-trained deep networks without access to gradients, loss functions, or original training data. By theoretically connecting local layer-wise reconstruction to global predictive error, we established *Subset Fidelity* as a rigorous proxy for component importance. Our empirical analysis reveals a fundamental property of modern networks: predictive performance is concentrated in sparse HIFI substructures that are robust to noise and identifiable via synthetic data. Leveraging this insight, we proposed MODHIFI, a unified framework for model modification. Unlike prior architecture-specific heuristics, MODHIFI is domain-agnostic, effectively handling both structured pruning and classwise unlearning across CNNs and Transformers. Crucially, our method is designed for the regime of *distributional access*, making it uniquely suited for modern deployments where privacy or scale necessitates the use of synthetic data.

**Limitations and Future Work** Our theoretical bounds in Theorem 3.6 rely on the local Lipschitz continuity of the network. While we demonstrate that this property holds for well-trained models (including Transformers on bounded domains), it is not guaranteed at initialization. This suggests that the emergence of High-Fidelity components is a consequence of the training dynamics. In this work, we use the expected square loss as a measure of distributional similarity, and we leave for future work the exploration of other metrics of distributional similarity, like the TV distance or Wasserstein metric.

## Acknowledgments and Disclosure of Funding

We, the authors, gratefully acknowledge AMD for its support. The authors thank Ramaswamy Govindarajan (IISc) and Prakash Raghavendra (AMD) for their insight and assistance in this work. The authors are also grateful to the reviewers of this work for their valuable feedback, which has significantly improved the content.

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

# APPENDIX

This appendix and their takeaways are summarized below for ease of navigation:

1. Appendix A contains additional related work and description of the gaps in existing literature addressed in our work.

2. Appendix B contains proofs and discussions not presented in the main body. In particular, we present the following:

   - In Appendix B.1, we provide the proof for Theorem 3.6. We also state and justify the assumption used towards proving the Theorem.
   - In Appendix B.2, we prove the properties of Subset Fidelity stated in Lemma 3.2.
   - In Appendix B.4, we prove the optimality of the naive algorithm in selecting the $k$-MFS Optimal set. While the assumption of uncorrelated features might not hold under practical scenarios, this result provides an indication that the method could result in effective identification of critical model components in practical settings. Our experiments in Section 5 and Appendix C practically demonstrate the empirical efficacy of the methodology.

3. Appendix C contains additional experimental validation that make the following points:

   - In Appendix C.1 we validate the existence of HiFi sets.
     - In Appendix C.1.1, we show the results of the full Monte-Carlo experiments on more models and datasets to strengthen our answer for Question (Q1).
     - In Appendix C.1.2, we conduct counterfactual experiments to validate if the sets computed by our proposed method are disproportionately responsible for predictive performance. **This emphasizes the effectiveness of Subset Fidelity in addition to our theoretical results in Theorem 3.9**.
     - In Appendix C.1.3, we empirically discuss the sensitivity of the estimation of $\boldsymbol{Q}^c$. This ablation study shows that the Subset Fidelity score is robust to the number of samples used for estimation. Among the datasets used, we see that we require at most 200 synthetic samples per class for accurate estimation.
   - In Appendix C.3, we study the effect of quality of synthetic samples on our proposed method. We find that higher quality data leads to an improved sparsity accuracy tradeoff.
   - In Appendix C.4 we provide pruning results on additional datasets strengthen our validation of Question (Q3).
     - In Appendix C.4.1 we show that each sub-module of our model modification is critical towards successful model modification.
     - In Appendix C.4.2 we compare the pruned ImageNet models of ModHiFi-P against SoTA data-free structured pruning DFPC [56] to compare layer-wise sparsity to see where is improved speedup coming from.
     - In Appendix C.4.3, we justify the appropriateness of our baselines for the LLM pruning baselines.
   - In Appendix C.5 we provide additional unlearning experiments with different models to strengthen our validation of Question (Q4) and discuss unlearning with finetuning. We validate that ModHifi-U performs competitively without finetuning against baselines. Moreover, with very few epochs of finetuning over synthetically generated samples, we achieve complete unlearning, as opposed to our baselines who fully-finetune on entire training set.
   - In Appendix C.6, we discuss implementation and compute platform details and additional timing measurements, to improve replicability of our empirical results.
   - In Appendix C.7 we discuss hyperparameters used for training, to improve replicability of our empirical results.

4. Appendix D contains additional algorithmic details including

   - Appendix D.1 clarifies of the role of Lipschitz Constants in our algorithms since they are critical to Theorem 3.6.
   - Appendix D.2 presents practical details as to how we implement fidelity estimation in our model modification algorithms.

- Appendix D.3 presents the computational complexity of estimating fidelity. The fidelity is linear in number of layers as opposed to the component identification module of SoTA in data-free structured pruning, which is quadratic in the number of layers.

# A  Related Work

We present a short literature review in Section 2. In this section, we discuss recent related work on structured pruning and unlearning not discussed in the main body.

**Structured Pruning**  Structured pruning has been widely researched, with a wide variety of methods proposed for it [27]. Unlike unstructured pruning, which sparsifies the weight matrices without changing the architecture of the model [6, 15, 16, 38, 73], structured pruning enables immediate improvements in real-world performance measures such as inference time and memory footprint without requiring specialized hardware or software [27, 51, 59]. A variety of methods have been proposed for structured pruning of convolutional networks, including using norms of weight tensors [39], directional derivative scores [51, 52, 66], feature map ranks [41, 70], coresets [4, 40, 77], discriminative ability of filters [42, 54], and reconstruction error [11, 67, 90]. However, modern neural networks possess complex interconnections, making them difficult to structurally compress [13, 43, 56], for which some recent algorithms have been proposed that use gradient information [43] or bounds on the reconstruction error [56, 90]. Moreover, pruning without access to either the training data or the loss function is an increasingly important area of research, for which some works have been proposed that use the discriminative ability of filters as a saliency [42, 54]. However, none of these works address the problem of pruning large language models.

Pruning of Large Language Models (LLMs) has garnered significant interest in recent years [94]. A variety of unstructured pruning methods have been proposed, such as [17, 71]. However, these methods do not provide direct improvements on inference time and memory footprint. Thus, the problem of pruning models with structural interconnections has naturally been applied to pruning LLMs as well, in works such as [2, 46, 47, 84]. A key drawback of these works is that most are not applicable to CNNs or other kinds of models. Our work proposes a unified framework for both pruning models with complex interconnections, including transformers and ResNets, as well as classwise unlearning.

**Classwise Unlearning**  Machine unlearning has gained significant interest in recent years, both for data privacy concerns as well as connections to continual learning [7, 29, 57, 81]. Machine unlearning is typically categorized into exact and approximate unlearning [86]. Exact unlearning involves training models from scratch without the forget data (the data to be forgotten), or by training modules or experts on subsets of data [85, 87]. Approximate unlearning, on the other hand, refers to techniques that approximate exact unlearning via various approaches [30, 86]. Machine unlearning can be further classified into sample unlearning (wherein individual samples or random subsets of samples are unlearned)[64, 78] or classwise unlearning (where classes or concepts are unlearned) [18, 24]. In this work, we focus on classwise unlearning.

A variety of approaches have been proposed for classwise unlearning [24]. Popular methods include fine-tuning the model without data from the forget class [20, 82], gradient ascent on the forget set [24, 75], distillation-based approaches [37], and influence function based methods [30]. More recent work studies using sparsity for machine unlearning, such as [32], which first sparsifies the model, and then applies a fine-tuning-based unlearning algorithm, or [53, 80], which identify class-discriminative filters in CNNs, and removes them for unlearning. Two key drawbacks of prior art, however, are: first they exclusively address classwise unlearning, and do not address wider problems of model modification. Second, all prior art assumes access to the original training data. Our proposed approach for classwise unlearning differs from prior art because it only requires synthetic class data, uses a variety of granularities for sparsity in unlearning, and is part of a unified approach to model modification.

# B  Proofs

In this section, we restate the formal statements made in the main body of the paper and present the proofs omitted in the main body. We follow the notation defined in Section 2.

## B.1 Proof of Theorem 3.6

We now provide a proof of Theorem 3.6. We first state the properties of the normalization layer and provide empirical evidence to justify their validity, followed by a restatement of the theorem and its proof.

**Definition B.1** (RMSNorm and LayerNorm). Consider the $l$-th layer parameterized by $\boldsymbol{\gamma}^l, \boldsymbol{\beta}^l \in \mathbb{R}^d$. For an input $\boldsymbol{\phi}(\boldsymbol{x}) \in \mathbb{R}^d$, the output $\mathbf{y} \in \mathbb{R}^d$ is defined as:

$$\mathrm{NM}(\boldsymbol{\phi}(\boldsymbol{x})) = \boldsymbol{\gamma}^l \odot \frac{\mathbf{z}}{\|\mathbf{z}\|_2} + \boldsymbol{\beta}^l, \quad \text{where } \mathbf{z} = \mathbf{M}\boldsymbol{\phi}(\boldsymbol{x}).$$

Here, $\odot$ denotes the Hadamard product. For RMSNorm, $\mathbf{M} = \mathbf{I}_d$. For LayerNorm, $\mathbf{M} = \mathbf{I}_d - \frac{1}{d}\mathbf{1}_d\mathbf{1}_d^\top$ (the centering matrix).

Normalization layers are not globally Lipschitz continuous due to the singularity at zero. However, they are locally Lipschitz on domains bounded away from the origin.

**Definition B.2.** A function $f : \mathbb{R}^m \to \mathbb{R}^n$ is Lipschitz continuous in its domain if there exists a positive scalar constant L such that

$$\|f(\boldsymbol{x}) - f(\boldsymbol{y})\|_2 \le L\|\boldsymbol{x} - \boldsymbol{y}\|_2 \quad \forall \boldsymbol{x}, \boldsymbol{y} \in \mathbb{R}^m$$

for all $\boldsymbol{x}, \boldsymbol{y}$ in the domain of $f$.

**Lemma B.3.** *Let* $\mathcal{X}_r = \{\mathbf{x} \in \mathbb{R}^d \mid \|\mathbf{x}\|_2 \ge r > 0\}$. *Define the map* $f : \mathcal{X}_r \to \mathbb{S}^{d-1}$ *as* $f(\mathbf{x}) = \frac{\mathbf{x}}{\|\mathbf{x}\|_2}$. *Then $f$ is Lipschitz continuous on $\mathcal{X}_r$ with constant $L_f = 1/r$. That is,*

$$\|f(\boldsymbol{x}) - f(\boldsymbol{y})\|_2 \le \frac{1}{r}\|\boldsymbol{x} - \boldsymbol{y}\|$$

*Proof.* For any $\mathbf{x}, \mathbf{y} \in \mathcal{X}_r$,

$$\|f(\boldsymbol{x}) - f(\boldsymbol{y})\|_2^2 = \left\| \frac{\boldsymbol{x}}{\|\boldsymbol{x}\|} - \frac{\boldsymbol{y}}{\|\boldsymbol{y}\|} \right\|_2^2$$
$$= 2 - 2\frac{\boldsymbol{x}^\top \boldsymbol{y}}{\|\boldsymbol{x}\|\|\boldsymbol{y}\|}$$

Simultaneously,

$$\|\mathbf{x} - \mathbf{y}\|_2^2 = \|\mathbf{x}\|^2 + \|\mathbf{y}\|^2 - 2\mathbf{x}^\top \mathbf{y}$$
$$= \|\mathbf{x}\|\|\mathbf{y}\| \left( \frac{\|\mathbf{x}\|}{\|\mathbf{y}\|} + \frac{\|\mathbf{y}\|}{\|\mathbf{x}\|} - 2\frac{\mathbf{x}^\top \mathbf{y}}{\|\mathbf{x}\|\|\mathbf{y}\|} \right).$$

Using the AM-GM inequality $a + 1/a \ge 2$ for $a > 0$, and noting that $\|\mathbf{x}\|, \|\mathbf{y}\| \ge r$:

$$\|\mathbf{x} - \mathbf{y}\|_2^2 \ge \|\mathbf{x}\|\|\mathbf{y}\| \left( 2 - 2\frac{\mathbf{x}^\top \mathbf{y}}{\|\mathbf{x}\|\|\mathbf{y}\|} \right) = \|\mathbf{x}\|\|\mathbf{y}\| \, \|f(\mathbf{x}) - f(\mathbf{y})\|_2^2 \ge r^2 \|f(\mathbf{x}) - f(\mathbf{y})\|_2^2.$$

Rearranging terms completes the proof. □

Using Lemma B.3, we can show that the operation performed by normalization layers is Lipschitz continuous in Corollary B.4.

**Corollary B.4.** *Let the input to the $l$-th normalization layer satisfy $\|\mathbf{M}\boldsymbol{\Phi}(\boldsymbol{x})\|_2 \ge r > 0$ for all $\boldsymbol{x}$. Then, the normalization layer satisfies*

$$\|\mathrm{NM}(\boldsymbol{\phi}(\boldsymbol{x})) - \mathrm{NM}(\boldsymbol{\phi}(\boldsymbol{y}))\| \le \frac{\|\boldsymbol{\gamma}^l\|_\infty}{r}\|\mathbf{M}\|_2\|\boldsymbol{\phi}(\boldsymbol{x}) - \boldsymbol{\phi}(\boldsymbol{y})\|.$$

*Note that $\|\mathbf{M}\|_2 = 1$ for both RMSNorm and LayerNorm.*

*Proof.* Let $\mathbf{u}(\mathbf{x}) = \mathbf{M}\mathbf{x}$. Then $\|\mathrm{NM}(\mathbf{x}) - \mathrm{NM}(\mathbf{y})\|_2 = \|\boldsymbol{\gamma}^l \odot \left( \frac{\mathbf{u}(\mathbf{x})}{\|\mathbf{u}(\boldsymbol{x})\|} - \frac{\mathbf{u}(\mathbf{y})}{\|\mathbf{u}(\boldsymbol{y})\|} \right) \|_2 \le \|\boldsymbol{\gamma}^l\|_\infty \frac{1}{r}\|\mathbf{M}(\mathbf{x} - \mathbf{y})\|_2$ by applying Lemma B.3 to complete the proof. □

While the lower bound assumption $\|\mathbf{z}\| \geq r$ is technically not guaranteed for all $\mathbf{x} \in \mathbb{R}^d$, we empirically verify that for trained networks, activation norms are strictly bounded away from zero. This validates the local Lipschitz property in the region of interest. We show the layer-wise minimum norm of the pre-LayerNorm representations in Figure 3, estimated on 100 samples from the Alpaca dataset. For various models, we observe the lower bound to be between 0.2 and 60. For clarity of exposition, we only show the layers with the largest and smallest values, along with 5 randomly selected layers. Code for generating these plots can be found in Appendix C. We also observe that this value tends to increase for layers deeper in the network, and leave the utilization of this observation to future work.

### B.1.1 Main Proof

We first state a well-known fact about Lipschitz functions. We then restate and prove Theorem 3.6.

**Fact 1.** *A function $f = f^L \circ f^{L-1} \circ \ldots \circ f^1$ where each $f^i$ is Lipschitz continuous with Lipschitz constant $L^i$, is Lipschitz continuous with Lipschitz constant $\prod_{i=1}^{L} L^i$.*

**Theorem 3.6** (Local to Global). *Consider a network $N_\theta$ as defined in Section 2.1. Let $\mathbf{M}^l$ be a mask modifying parameters at layer $l$, and let $\mathbf{m}_c^l$ be the mask vector for output channel $c$. Assume there exist scalars $r^\ell > 0$ for all layers $\ell > l$ such that $\|\mathbf{\Phi}_c^\ell(\mathrm{X})\|_F \geq r^\ell$ almost surely. Then,*

$$\mathbb{E}\left[\|N_\theta(\mathrm{X}) - N_{\theta \odot \mathbf{M}^l}(\mathrm{X})\|^2\right] \leq \mathcal{O}\left(\sum_{c=1}^{c_{out}^l} \mathbb{E}\left[\|\mathbf{Y}_c^l(\mathrm{X}) - \sum_{i \in C} m_{ci}^l \mathbf{A}_{ci}^l(\mathrm{X})\|^2\right]\right) \tag{2}$$

*Proof.* Consider a network as defined in Section 2.1. Let $N_\theta = f^L \circ \ldots \circ f^l \circ f^{l-1:1}$ where $f^{l-1:1} = f^{l-1} \circ \ldots \circ f^1$. Under standard assumptions on the smoothness of activations [90], each layer $f^l$ is Lipschitz continuous with Lipschitz constant $L_f^l$. From Fact 1,

$$\mathbb{E}\left[\|N_\theta(\mathrm{X}) - N_{\theta \odot \mathbf{M}^l}(\mathrm{X})\|^2\right] \leq (\prod_{\ell > l}^{L} L_f^\ell) \sum_{c=1}^{c_{out}^l} \mathbb{E}[\|\mathbf{Y}^l(\mathrm{X}) - \sum_i m_{ci} \mathbf{A}_{ci}(\mathrm{X})\|]^2$$

By taking an upper bound on the Lipschitz constants of each layer in the composition, we see that the subnetwork after layer $l$ has a Lipschitz constant of at least $C^l = \prod_{\ell > l}^{L} L_f^\ell$. Where, for convolution-based networks,

$$C_l = \left(\max_i \frac{\gamma_i^l}{\sigma_i^l}\right) \eta^{L-l} \prod_{\ell > l} \|\mathcal{W}^\ell\|_2 \cdot \max_i \frac{|\gamma_i^\ell|}{\sigma_i^\ell}$$

and for transformer models,

$$C_l = \eta^{L-l} \prod_{\ell > l} \|\mathcal{W}^\ell\|_2 \cdot \max_i \frac{|\gamma_i^\ell|}{r^\ell}$$

The expected squared error at the final output is:

$$\mathbb{E}[\|N_\theta(\mathrm{X}) - N_{\theta \odot \mathbf{M}}(\mathrm{X})\|^2] \leq C_l^2 \, \mathbb{E}[\|\mathbf{Y}^l(\mathrm{X}) - \tilde{Y}^l(\mathbf{X})\|^2].$$

We decompose the layer output by channels $c \in [c_{out}^l]$. The masked output for channel $c$ is $\tilde{\mathbf{Y}}_c^l = \sum_i m_{ci} \mathbf{A}_{ci}^l$, where $m_{ci} \in \{0, 1\}$ are entries of $\mathbf{M}^l$.

$$\mathbb{E}[\|\mathbf{Y}^l(\mathrm{X}) - \tilde{\mathbf{Y}}^l(\mathrm{X})\|^2] = \sum_{c=1}^{c_{out}^l} \mathbb{E}\left[\left\|\sum_{i=1}^{c_{in}} \mathbf{A}_{ci}^l(\mathrm{X}) - \sum_{i=1}^{c_{in}} m_{ci} \mathbf{A}_{ci}^l(\mathrm{X})\right\|^2\right]$$

$$= \sum_{c=1}^{c_{out}^l} \mathbb{E}\left[\left\|\sum_{i=1}^{c_{in}} (1 - m_{ci}) \mathbf{A}_{ci}^l(\mathrm{X})\right\|^2\right].$$

Let $\mathbf{v}_c = \mathbf{1} - \mathbf{m}_c$ be the indicator vector of removed components. Expanding the squared norm:

$$\mathbb{E}\left[\left\|\sum_i v_{ci}\mathbf{A}_{ci}^l(\mathrm{X})\right\|^2\right] = \mathbb{E}\left[\sum_i\sum_j v_{ci}v_{cj}\langle\mathbf{A}_{ci}^l(\mathrm{X}),\mathbf{A}_{cj}^l(\mathrm{X})\rangle\right]$$
$$= \sum_{i,j} v_{ci}(\mathbf{Q}_c^l)_{ij}v_{cj} = \mathbf{v}_c^\top\mathbf{Q}_c^l\mathbf{v}_c.$$

Substituting this back completes the proof. $\square$

## B.2 Proof of Lemma 3.2

In this section, we prove the properties of Subset Fidelity stated in Lemma 3.2. We restate the definition of fidelity score and state a proposition. We then restate the proposition and provide a proof.

**Definition 3.1** (Subset Fidelity). The *fidelity* of a subset of components $C \subseteq [c_{in}^l]$ in layer $l$ for output channel $c$ is defined as

$$\mathrm{FS}_c^l(C) := \max_{\boldsymbol{\delta}_c^l \in \mathbb{R}^{c_{in}^l}}\left(1 - \frac{\mathbb{E}\left[\|\mathbf{Y}_c^l(\mathrm{X}) - \sum_{i\in C}\delta_{ci}^l\mathbf{A}_{ci}^l(\mathrm{X})\|^2\right]}{\mathbb{E}\left[\|\mathbf{Y}_c^l(\mathrm{X})\|^2\right]}\right), \tag{1}$$

where $\boldsymbol{\delta}_c^l$ is the *compensation term*.

**Lemma 3.2** (Properties of Subset Fidelity). *For any subset $C \subseteq [c_{in}^l]$ in layer $l$, (**Boundedness**) $0 \le \mathrm{FS}_c^l(C) \le 1$ and (**Monotonicity**) If $D \subseteq C$, then $\mathrm{FS}_c^l(D) \le \mathrm{FS}_c^l(C)$.*

*Proof.* Let $\mathcal{L}(\boldsymbol{\delta}; C) := \mathbb{E}[\|\mathbf{Y}_c^l(\mathrm{X}) - \sum_{i\in C}\delta_i\mathbf{A}_{ci}^l(\mathrm{X})\|^2]$. The fidelity is $\mathrm{FS}_c^l(C) = 1 - \frac{\min_{\boldsymbol{\delta}}\mathcal{L}(\boldsymbol{\delta}; C)}{\mathbb{E}[\|\mathbf{Y}_c^l(\mathrm{X})\|^2]}$.

**1. Boundedness:** Since the norm is non-negative, $\mathcal{L}(\boldsymbol{\delta}; C) \ge 0 \implies \mathrm{FS} \le 1$. Selecting $\boldsymbol{\delta} = \mathbf{0}$ yields $\mathcal{L}(\mathbf{0}; C) = \mathbb{E}[\|\mathbf{Y}_c^l\|^2]$. Since the minimum is bounded by this value, the ratio is $\le 1$, so $\mathrm{FS} \ge 0$.

**2. Monotonicity:** Let $D \subset C$. The optimization for $D$ is equivalent to optimizing over $C$ with the constraint $\delta_i = 0 \,\forall i \in C \setminus D$. Since $D \subset C$, the feasible set for $D$ is a subset of the feasible set for $C$. Therefore, $\min_{\boldsymbol{\delta}}\mathcal{L}(\boldsymbol{\delta}; C) \le \min_{\boldsymbol{\delta}'}\mathcal{L}(\boldsymbol{\delta}'; D)$. A lower minimum error implies a higher fidelity score. Thus, $\mathrm{FS}_c^l(C) \ge \mathrm{FS}_c^l(D)$. $\square$

## B.3 Proof of Proposition 3.8

**Proposition 3.8** (Compensation and Singleton Fidelity). *For the $l_2$ reconstruction error, the optimal compensation term $\delta_c^\star$, which is the value at which the fidelity score is computed according to Equation (1) for a subset $C$, is given by,*

$$\delta_{ci}^{l\star}(C) = \begin{cases} 1 + ((\mathbf{Q}_c^l[C,C])^{-1})_i^\top \mathbf{Q}_c^l[C,\overline{C}]\mathbf{1}_{n-k} & \text{if } i \in C \\ 0 & \text{if } i \notin C \end{cases} \tag{FS}$$

*where $\mathbf{Q}_c^l \in \mathbb{R}^{c_{in}^l \times c_{in}^l}$ is the component similarity matrix (CSM) for channel $c$, with entries $(\mathbf{Q}_c^l)_{ij} = \mathbb{E}[\langle\mathbf{A}_{ci}^l(\mathrm{X}),\mathbf{A}_{cj}^l(\mathrm{X})\rangle]$. The singleton fidelity scores are:*

$$s_{ci}^l = \mathrm{FS}_c^l(\{i\}) = 1 - \frac{\mathbb{E}[\|\mathbf{Y}_c^l(\mathrm{X}) - \alpha_{ci}^l\mathbf{A}_{ci}^l(\mathrm{X})\|^2]}{\mathbb{E}[\|\mathbf{Y}_c^l(\mathrm{X})\|^2]}, \quad \alpha_{ci}^l = \frac{\mathbb{E}[\langle\mathbf{Y}_c^l(\mathrm{X}),\mathbf{A}_{ci}^l(\mathrm{X})\rangle]}{\mathbb{E}[\|\mathbf{A}_{ci}^l(\mathrm{X})\|^2]}. \tag{3}$$

*Proof.* Define the error $\mathbf{e}(\mathrm{X}) = \mathbf{Y}_c^l(\mathrm{X}) - \sum_{i\in C}\delta_i\mathbf{A}_{ci}^l(\mathrm{X})$. We minimize $J(\boldsymbol{\delta}) = \mathbb{E}[\|\mathbf{e}(\mathrm{X})\|^2]$. Recall that $\mathbf{Y}_c^l(\mathrm{X}) = \sum_{i=1}^{c_{in}}\mathbf{A}_{ci}^l(\mathrm{X})$. Let $\mathbf{u} = \mathbf{1} - \boldsymbol{\delta}$, where $\mathbf{u}$ is supported on the full set of indices but we constrain $\delta_i = 0$ (so $u_i = 1$) for $i \notin C$. The objective is:

$$J(\mathbf{u}) = \mathbb{E}\left[\left\|\sum_{i=1}^{c_{in}} u_i\mathbf{A}_{ci}^l(\mathrm{X})\right\|^2\right] = \mathbf{u}^\top\mathbf{Q}_c^l\mathbf{u}.$$

We partition indices into $C$ and $\overline{C}$. Decompose $\mathbf{u}$ as $[\mathbf{u}_C; \mathbf{u}_{\overline{C}}]$. The constraint $\delta_i = 0$ for $i \notin C$ implies $\mathbf{u}_{\overline{C}} = \mathbf{1}_{\overline{C}}$. We optimize with respect to $\mathbf{u}_C$:

$$J(\mathbf{u}_C) = \begin{bmatrix} \mathbf{u}_C^\top & \mathbf{1}_{\overline{C}}^\top \end{bmatrix} \begin{bmatrix} \mathbf{Q}_{CC} & \mathbf{Q}_{C\overline{C}} \\ \mathbf{Q}_{\overline{C}C} & \mathbf{Q}_{\overline{C}\overline{C}} \end{bmatrix} \begin{bmatrix} \mathbf{u}_C \\ \mathbf{1}_{\overline{C}} \end{bmatrix}$$
$$= \mathbf{u}_C^\top \mathbf{Q}_{CC}\mathbf{u}_C + 2\mathbf{u}_C^\top \mathbf{Q}_{C\overline{C}}\mathbf{1}_{\overline{C}} + \text{const.}$$

This is a convex quadratic function, whose optima can be computed by taking the gradient w.r.t $\mathbf{u}_C$ and setting to zero:

$$2\mathbf{Q}_{CC}\mathbf{u}_C + 2\mathbf{Q}_{C\overline{C}}\mathbf{1}_{\overline{C}} = 0 \implies \mathbf{u}_C^\star = -\mathbf{Q}_{CC}^{-1}\mathbf{Q}_{C\overline{C}}\mathbf{1}_{\overline{C}}.$$

Recalling $\boldsymbol{\delta}_C = \mathbf{1}_C - \mathbf{u}_C$, we obtain:

$$\boldsymbol{\delta}_C^\star = \mathbf{1}_C + \mathbf{Q}_{CC}^{-1}\mathbf{Q}_{C\overline{C}}\mathbf{1}_{\overline{C}}.$$

This matches Equation (3). $\qquad\square$

### B.4 Proof of Theorem 3.9

In this section, we prove the optimality of the naive algorithm in selecting the $k$-MFS Optimal set.

**Theorem 3.9.** *Consider output channel $c$ in the $l^{th}$ layer of a network described in Section 2.1. Let the $s_{ci}^l$ be defined according to Equation (3) and $S_c^{l\star}$ be defined according to Definition 3.5. Let $\hat{S}_c^l = \{i \mid s_{ci}^l \geq s_{(k)}\}$ where $s_{(k)}$ is the $k^{th}$ largest value of $\mathbf{s}_c^l$. Assuming that there are no ties, $|\hat{S}_c^l| = k$. If $\mathbb{E}[\langle \boldsymbol{A}_{ci}^l(X), \boldsymbol{A}_{cj}^l(X)\rangle] = 0 \ \forall i \neq j$, then $\hat{S}_c^l = S_c^{l\star}$.*

*Proof.* The assumption $\mathbb{E}[\langle \mathbf{A}_{ci}^l, \mathbf{A}_{cj}^l\rangle] = 0$ for $i \neq j$ implies that the Component Similarity Matrix $\mathbf{Q}_c^l$ is diagonal. Let $q_{ii} = (\mathbf{Q}_c^l)_{ii} = \mathbb{E}[\|\mathbf{A}_{ci}^l\|^2] \geq 0$. This implies that the component similarity matrix is diagonal. For any subset $S$, the optimal compensation $\boldsymbol{\delta}^\star$ for diagonal $\mathbf{Q}$ simplifies. The reconstruction error for subset $S$ is minimized when we perfectly reconstruct the components in $S$ (since they are orthogonal to components in $\overline{S}$). Thus, the residual error comes purely from the removed components $\overline{S}$:

$$\min_{\boldsymbol{\delta}} \mathbb{E}\left[\left\|\mathbf{Y}_c^l(X) - \sum_{i \in S}\delta_i \mathbf{A}_{ci}^l(X)\right\|^2\right] = \mathbb{E}\left[\left\|\sum_{j \in \overline{S}}\mathbf{A}_{cj}^l(X)\right\|^2\right] = \sum_{j \notin S} q_{jj}.$$

The Subset Fidelity is:

$$\mathrm{FS}_c^l(S) = 1 - \frac{\sum_{j \notin S}q_{jj}}{\sum_k q_{kk}} = \frac{\sum_{i \in S}q_{ii}}{\mathrm{Tr}(\mathbf{Q}_c^l)}.$$

Similarly, the singleton fidelity score for component $i$ is $s_{ci}^l = \frac{q_{ii}}{\mathrm{Tr}(\mathbf{Q}_c^l)}$. The optimization problem:

$$S^\star = \arg\max_{|S|=k} \mathrm{FS}_c^l(S) = \arg\max_{|S|=k} \sum_{i \in S} q_{ii}.$$

This linear objective is trivially maximized by selecting the $k$ indices with the largest $q_{ii}$ values. Since $s_{ci}^l \propto q_{ii}$, this is equivalent to selecting the top-$k$ singleton fidelity scores. $\qquad\square$

*Remark B.5.* While the assumption of uncorrelated features might not hold under practical scenarios, this result provides an indication that the method could result in effective identification of critical model components in practical settings. Our experiments in Section 5 and Appendix C practically demonstrate the empirical efficacy of the methodology.

## C  Additional Experiments

In this appendix we detail additional results and ablations.

1. We elaborate on the Monte Carlo simulations of Equation ($\kappa$-MFS) across multiple models, as well as the efficiency with which Subset Fidelity estimates this while being robust to data samples. We present noising and counterfactual results to demonstrate this.

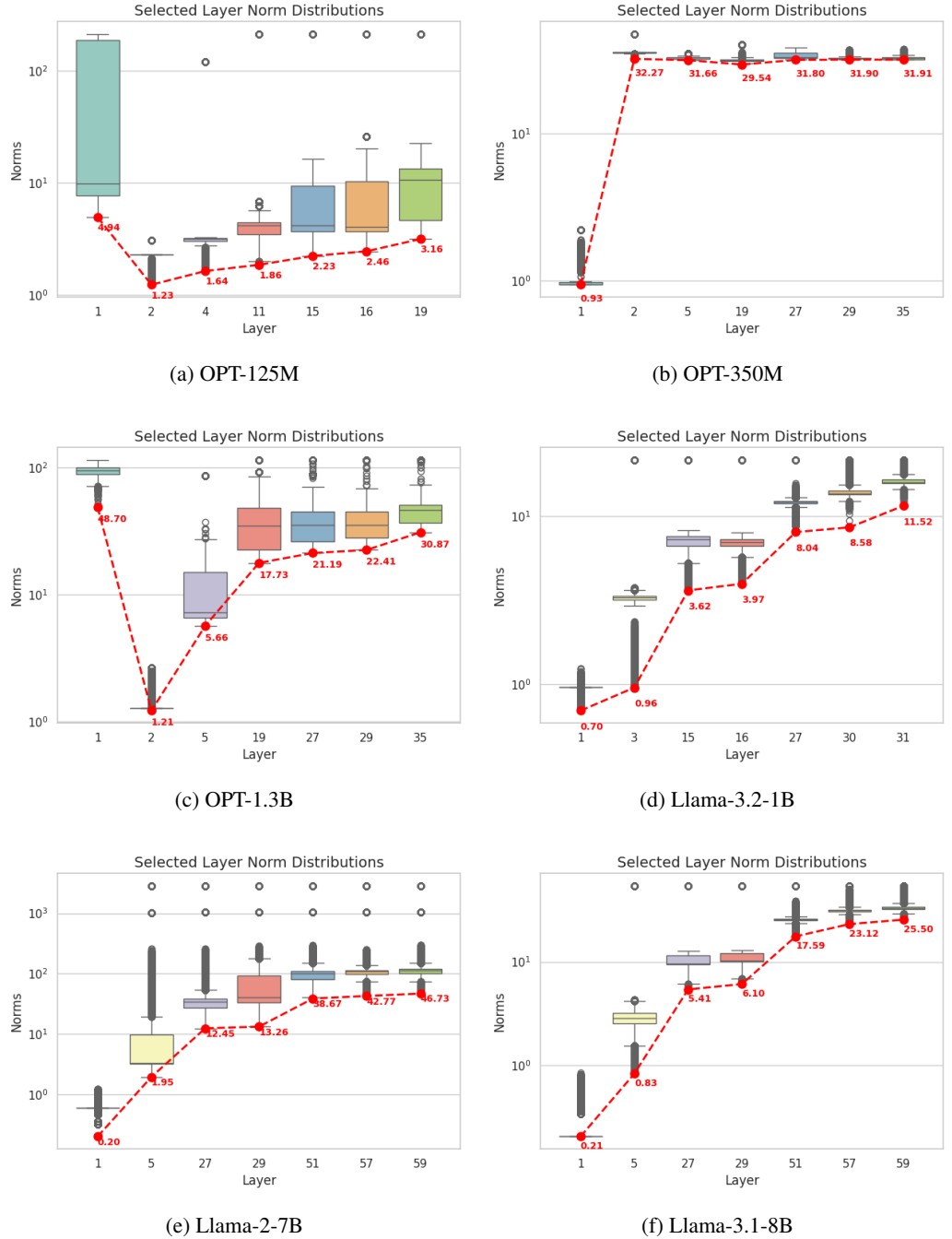

Figure 3: Boxplots for the distribution of norms of inputs to normalization layer. Minimum value indicated in Red, showing that $\frac{1}{r}$ is at most 5. Y-axis is log scale.

2. We discuss the synthetic samples used in our vision experiments and the effect of their quality on the algorithm.

3. We provide additional pruning and unlearning experiments for a variety of architectures.

4. We provide details of our compute platform, hyperparameters, and training procedure for our experiments.

Our code is available at https://github.com/DhruvaKashyap/modhifi.

### C.1 Validating Subset Fidelity and HɪFɪ Sets

#### C.1.1 Monte Carlo Experiments

In this section, we provide additional details regarding Figure 1 and demonstrate this behaviour across architectures.

Ideally, one would solve Equation (κ-MFS) exactly to compute those sets that have optimal reconstructive ability at different sizes. However, since enumerating across subsets is a combinatorial problem, we instead approximate a solution by randomly sampling 1000 sets of the given size and compute the maximum across these samples. *This will always provide a lower bound for the "true" curve*.

Solving Equation (κ-MFS) allows us to compute the optimal $(k, \eta)$-HɪFɪ sets, since this captures the relation between $\eta$ and $k$, i.e. the tradeoff between sparsity and accuracy. We observe that in many layers (at least 50% of the model), across multiple models, there are sets which contain at most 20% of components but have a subset fidelity of around $0.8$. We also observe that as the difficulty of the task increases (CIFAR10 to ImageNet), fewer layers exhibit this sparsity, validating the assertion that networks trained on harder problems are less overparameterized.

Figures 4 to 6 show this for a ResNet-50 trained on CIFAR10, CIFAR100, and ImageNet. Figure 7 shows this for a OPT-125m model using 128 samples of WikiText and 100 random subsets instead of a 1000. Due to the expensive nature of this experiment, we are forced to use only 100 random subsets for each size, leading to a noisier curve. However, it is clear to see that the general trend continues to hold for several layers, especially for the "down-projection" weight matrices, which are the focus of our pruning algorithm for LLMs.

#### C.1.2 Counterfactual study of HiFi sets

We compare the effect of removing HiFi components from a layer with the effect of removing a random subset of the same size. For a ResNet-50 model trained on CIFAR-10, when around 22% of HiFi components are removed, the accuracy drops by around 70%, whereas removing a random subset of the same size decreases the accuracy by 32%. Note that there is a roughly 1% decrease in accuracy when only 22% of the non-HiFi components are removed. This indicates that components classed as "High Fidelity" have a significantly higher impact on the model's predictive performance than those with lower fidelity scores.

#### C.1.3 Robustness of the Fidelity Score

In this section, we perform ablations on the number of samples required for estimating the fidelity score and show how it reacts to additive noise on the model's weights.

In Figures 8 and 9 we show how different data sizes affect different layers in a ResNet50 model trained on CIFAR10 and ImageNet, respectively. Each data size is selected over 3 random seeds, with error bars shown. For clarity, we show only a subset of layers and provide plots and code to generate them.

We observe that the values remain stable for 0.2%, 0.5%, and 2% of the data selected, indicating that the model is robust to the number of samples selected.

We also observe the effect of training in these graphs. In untrained models, almost all components have very small fidelity scores with a sharp increase for some values. This indicates that HiFi components are a function of training, with the well-trainedness of the network being a prerequisite for their presence

When investigating the effect of adding noise to the weights and its effect on accuracy and the fidelity score, we observe that adding zero mean noise of larger standard deviations, starting from 0.005 to 0.05, decreases the fidelity of components, with noisier weights behaving more like untrained models. We test this on a ResNet-50 trained on CIFAR10 and present the results in Figure 10. Again, we present only a random subset of the layers for clarity.

Table 5: Effect of data quality when pruning a ResNet-50 on CIFAR10

| FID | Diffusion Steps | Accuracy | FLOP Reduction | Param Reduction |
|------|------|------|------|------|
| 85.80 | 4 | 86.27 | 2.63x | 2.66x |
| 35.58 | 5 | 90.75 | 2.78x | 2.78x |
| 14.42 | 6 | 90.39 | 3.50x | 3.60x |

## C.2 Constants in Theorem 3.6

In this section, we provided worst case and average case estimates of the constant $C_l$ in Theorem 3.6. In Figure 11a, we plot the constants obtained in the proof of Theorem 3.6 in Appendix B.1 for a ResNet-50 trained on ImageNet, and observe that the values can be very large (38 orders of magnitude). However, it is important to note that these are worst case guarantees, and that these constants are much smaller in practice. In Figure 11b, we compute the ratio between the global error, $\mathbb{E}\left[\|\boldsymbol{y}(\mathrm{X}) - \boldsymbol{y}(\mathrm{X}; \boldsymbol{M}^l)\|^2\right]$ and the local error, $\sum_{c=1}^{c_{out}^l} \mathbb{E}\left[\left\|\boldsymbol{Y}_c^l(\mathrm{X}) - \sum_{i \in C} m_{ci}^l \boldsymbol{A}_{ci}^l(\mathrm{X})\right\|^2\right]$ and observe that these values are indeed much smaller (10-50) for random values of $\boldsymbol{M}^l$ for the expected square loss.

## C.3 Discussion on the synthetic samples used in the experiments

We describe the synthetic datasets used in our vision experiments to simulate distributional access. Randomly selected example images are provided in Figure 12. For NLP tasks, we use WikiText and Alpaca datasets [48, 74] which are standard in this field.

### C.3.1 CIFAR5M

For experiments with the CIFAR10 dataset, we use CIFAR5M, a dataset containing 6 million synthetic CIFAR-10-like images sampled from a Diffusion model and labeled by a Big-Transfer model [55], which we randomly sample 10,000 samples from each of the 10 classes to create our dataset. This dataset has an FID [26] of 15.95 with respect to the CIFAR10 training set. This dataset is obtained from here.

### C.3.2 CIFAR100-DDPM

For experiments with the CIFAR100 dataset, we use CIFAR100-DDPM [23], which we randomly downsample to contain 1,000 samples from each of the 100 classes. This dataset has an FID of 4.74 with respect to the CIFAR100 training set. We randomly sample 1,000 samples from each of the 100 classes to create our dataset. This dataset is obtained from here.

### C.3.3 Effect of Data Quality

To study the effect of data quality on the performance of our algorithm in vision tasks, we apply the pruning algorithm using synthetic datasets based on CIFAR10 generated with different FIDs. We use a diffusion model [33] to generate 3 datasets of differing quality by changing the number of diffusion steps (4,5, and 6). We report the results of our pruning algorithm with different quality datasets in Table 5. We observe that higher quality data leads to an improved sparsity - accuracy tradeoff.

## C.4 Additional Pruning Experiments

We present additional pruning experiments in Tables 6 and 7.

### C.4.1 Ablation of weight compensation and BatchNorm correction

In this section, we perform ablations for each component of our pruning algorithm, simple pruning, correcting batch norm statistics and weight compensation. We report our results for pruning ResNet-50 on CIFAR 10 in Table 8. We observe that each component allows for a better accuracy sparsity trade-off.

Table 6: Comparison of ResNet-50 pruning for CIFAR10 and CIFAR100. ST = *Synthetic Training*, i.e. training using synthetic samples.

| Dataset | Algorithm | Accuracy | FLOP Reduction | Param Reduction |
|---------|-----------|----------|----------------|-----------------|
| CIFAR10 | Unpruned | 94.99 | 1x | 1x |
| | DFPC | 90.25 | 1.46x | 2.07x |
| | $L_2$ | 15.91 | 4.07x | 4.71x |
| | $L_2$ w/ ST | 90.12 | 4.07x | 4.71x |
| | **Ours** | **91.02** | **4.07x** | **5.36x** |
| CIFAR100 | Unpruned | 78.85 | 1x | 1x |
| | DFPC | 70.31 | 1.27x | 1.22x |
| | $L_2$ | 16.77 | 1.93x | 1.40x |
| | $L_2$ w/ ST | **73.83** | 1.93x | **1.40x** |
| | **Ours** | 70.93 | **1.93x** | 1.38x |

Table 7: Comparison of ResNet-101/VGG-19 pruning on CIFAR10 and CIFAR100. ST = *Synthetic Training*, i.e. training using synthetic samples.

| Dataset | Model | Algorithm | Accuracy | FLOP Reduction | Param Reduction |
|---------|-------|-----------|----------|----------------|-----------------|
| CIFAR-100 | VGG19 | Unpruned | 72.02 | 1x | 1x |
| | | DFPC | 70.10 | 1.26x | 1.50x |
| | | $L_2$ | 56.46 | 1.50x | 2.40x |
| | | $L_2$ w/ ST | **72.42** | 1.50x | **2.40x** |
| | | **Ours** | 70.26 | **1.51x** | 2.31x |
| CIFAR10 | ResNet-101 | Unpruned | 95.09 | 1x | 1x |
| | | DFPC | 89.80 | 1.53x | 1.84x |
| | | $L_2$ w/ ST | 90.49 | 4.20 | **5.29x** |
| | | **Ours** | **91.20** | **4.21x** | 4.79x |
| | VGG19 | Unpruned | 93.50 | 1x | 1x |
| | | DFPC | 90.25 | 1.46x | 2.07x |
| | | $L_2$ w/ ST | 89.23 | 2.39x | **9.19x** |
| | | **Ours** | **91.80** | **2.39x** | 5.52x |

## C.4.2 Final ImageNet Pruned Model

In Figures 13 and 14 we compare the final pruned models for ResNet-50 on ImageNet with DFPC [56]. We observe that our pruning algorithm removes more channels in later coupled channels than DFPC leading to higher gains in sparsity.

## C.4.3 Baseline selection for LLM Pruning

We choose ShortGPT [47] and SliceGPT [2] as baselines against which we compare ModHiFi. We do so for two broad reasons: all three methods together represent three different granularities for conducting structured pruning for LLMs, and both ShortGPT and SliceGPT are the state-of-the-art within their respective lanes.

The three different granularities are

1. Layer pruning: Entire layers (i.e. transformer decoder blocks) are removed from the network. This is viable since transformers are constant width networks, i.e., there are no architectural restrictions to the ordering or number of layers. ShortGPT falls within this granularity.

Table 8: Ablation of different components

| BatchNorm | Compensation | Accuracy | FLOP Reduction | Param Reduction |
|-----------|--------------|----------|----------------|-----------------|
| No | No | 93.37 | 1.61x | 1.53x |
| No | Yes | 93.49 | 2.21x | 2.17x |
| Yes | No | 93.17 | 2.53x | 2.39x |
| Yes | Yes | 93.76 | 3.22x | 3.30x |

2. Embedding pruning: The width of the network (i.e. the embedding dimension) is pruned at a uniform rate across the entire network. This entails a form of feature selection: along with weight matrix pruning, one also has to prune the corresponding dimensions from the feature matrix being fed into every layer. SliceGPT falls within this granularity.

3. Hidden dimension pruning: Here, the number of layers and the width of the embedding are left unchanged. Instead, one prunes the hidden dimensions within the modules that constitute a transformer decoder block. ModHiFi falls within this granularity.

We would like to emphasize that both SliceGPT and ShortGPT are designed to operate on Transformer models, and as such are able to leverage specifics of the architecture to their advantage. In return for this specificity, however, they trade off the ability to generalize to CNNs, something that ModHiFi does with ease due to its architecture-agnostic nature; the only assumption made by the Fidelity Score is that the components being scored belong to linear layers.

## C.5 Additional Unlearning Experiments

We report additional experiments in Table 9 on class unlearning on different architectures. For VGG-19 networks, we remove the HiFi channels for the forget class of the last 12 convolution layers. We also compare our work with DisCEdit-U from [53] wherein we remove discriminative components from the last 8 convolutional layers. We use a custom implementation of the algorithm for our VGG19 and ResNet50 models for CIFAR10, as those models are unavailable in the codebase of [53].

We also compare our work with DisCEdit-U on ResNet50 trained on CIFAR10 as well, which we present in Table 10

We show that our unlearning method achieves similar or superior performance to that of [53] without fine-tuning. Moreover, unlike [53], our approach uses only *synthetic samples*, showing the efficacy of our work in classwise unlearning, even in the absence of training data.

**Unlearning with finetuning**   Here we compare our method with 3 additional epochs of finetuning on synthetic samples of the remaining class data. Although this setup does not fall into the setup of the work since we do not assume access to the loss function, we provide these results to indicate that even using very few synthetic samples we can perform perfect unlearning. We present these results in Table 11, where we observe almost perfect unlearning for both ResNet-50 and Swin-Transformers.

**Unlearning with baseline budgets**   In this section, we compare our method when allowing for the same amount of finetuning as [32], with both synthetic data and training data access. While this violates our assumptions about loss function and training data access, we present these results to provide a fair comparison of our algorithm when run within the same constraints as our baselines. Our results can be found in Table 12 for the Swin Transformer.

## C.6 Compute Platform

**Implementation Details**   We implement our proposed methods in PyTorch [58] and use Hugging-face's `transformers` [83] for LLM implementations.

**Inference time measurements**   We follow the inference time measurement setting of [56, 66]. Inference time is the time taken for a model to compute the forward pass for an input and does not account for loading data into memory. We compute the inference time for a batch of 640 random tensors for GPU and 64 for CPU. 100 iterations are used for warm up, after which the inference

Table 9: Class unlearning on CIFAR10 for VGG19

| Model | Algorithm | Forget Accuracy | Remain Accuracy |
|-------|-----------|-----------------|-----------------|
| VGG19 | - | 93.50 | 93.50 |
| | DisCEdit-U [53] | 2.39 | **84.2** |
| | Ours | **0.86** | 77.85 |

Table 10: Class unlearning on CIFAR10 for ResNet50

| Model | Algorithm | Forget Accuracy | Remain Accuracy |
|-------|-----------|-----------------|-----------------|
| ResNet50 | - | 94.99 | 94.99 |
| | DisCEdit-U [53] | 3.2 | 91.6 |
| | Ours | **0.2** | **92.98** |

Table 11: Class unlearning with 3 epochs of finetuning on synthetic samples

| Model | Remain Accuracy | Forget Accuracy |
|-------|-----------------|-----------------|
| ResNet-50 | 93.1 | 0 |
| Swin-T | 83.6 | 0.1 |

time is averaged over the next 1000 forward passes. We compute CPU and GPU measurements on a machine whose specifications can be found in Appendix C.6.

**JIT Compilation**    We present inference time numbers with JIT compilation on Pytorch [58].

**Hardware**    Table 13 details the hardware we use to conduct our experiments. Values in (*) indicate reported values obtained from https://www.amd.com/en/products/accelerators/instinct/mi200/mi210.html. This machine runs Ubuntu 22.04.3 LTS with kernel 6.8.0-40-generic with the hardware in Table 13. Our software stack comprises of Python 3.12.8, PyTorch 2.5.1 built for ROCm 6.2, and torchvision version 0.20.1 built for ROCm 6.2.

Inference times are measured on a machine running Ubuntu 20.04.1 LTS with kernel 5.15.0-91-generic on the hardware specified in Table 14. The software stack used for inference consists of Python 3.12.8, PyTorch 2.5.1, and Torchvision 0.20.1 for CUDA 12.3.

### C.6.1    Module-level Time Consumption

In this section, we break down the time each component of our algorithm takes. For 2000 samples batched into batches of size 64, when running the algorithm on a ResNet-50:

- Computation of fidelity scores takes between 32GB to 51GB of VRAM, and between 2 minutes to 5 minutes, on 1 GPU of machine 13, across data from CIFAR10, CIFAR100, and ImageNet.

- Computing $\delta_c^\star$ across 4 GPUs using an average of 60GB per GPU takes 60 minutes for CIFAR10/100, and 90 minutes for ImageNet, averaging to roughly 1 minute per layer.

### C.7    Hyperparameters and Training Procedure

### C.7.1    Hyperparameters for Experiments

We typically set the percentile of removed components to be between 0.01 to 0.2. We randomly select 2% of our synthetic samples to select data for vision tasks and select 128 samples for NLP tasks.

Table 12: Class unlearning with 10 epochs of finetuning

| Approach | Dataset | Forget Accuracy | Remain Accuracy |
|----------|---------|-----------------|-----------------|
| Jia et al. [32] | CIFAR10 Train | 1.20 | 90.69 |
| **Ours** | CIFAR10 Synthetic | 0.37 | 84.63 |
| **Ours** | CIFAR10 Train | **0.00** | **91.1** |

Table 13: Specifications of GPU hardware used for computation

| | |
|---|---|
| CPU Model Name | AMD EPYC 9654 96-Core Processor |
| CPU(s) | 192 |
| Thread(s) per core | 1 |
| Core(s) per socket | 96 |
| Socket(s) | 2 |
| NUMA node(s) | 2 |
| CPU MHz(Max) | 3707.8120 |
| L1d & L1i cache | 6 MiB |
| L2 cache | 192 MiB |
| L3 cache | 768 MiB |
| RAM | 1.48 TiB (DDR5, 4800 MT/s) |
| GPU Model name | Instinct MI210 |
| GPU(s) | 4 |
| GPU Architecture | AMD Aldebaran |
| Dedicated Memory Size(per GPU) | 64 GB |
| ROCm Version | 6.0.2 |
| Peak FP32 Performance* | 22.6 TFLOPs |
| Peak FP64 Performance* | 22.6 TFLOPs |
| Memory Clock* | 1.6 GHz |
| Peak Memory Bandwidth* | 1.6 TB/s |

Table 14: Specifications of GPU and CPU hardware used for computing inference time

| | |
|---|---|
| CPU Model Name | Intel(R) Xeon(R) Silver 4216 CPU @ 2.10GHz |
| CPU(s) | 64 |
| Thread(s) per core | 2 |
| Core(s) per socket | 16 |
| Socket(s) | 2 |
| NUMA node(s) | 2 |
| CPU MHz(Max) | 3200 |
| L1d & L1i cache | 1 MiB |
| L2 cache | 32 MiB |
| L3 cache | 44 MiB |
| RAM | 62.53 GiB (DDR4, 2666 MT/s) |
| GPU Model name | NVIDIA GeForce RTX 2080 Ti |
| CUDA version | 12.3 |
| GPU(s) | 8 |
| GPU Architecture | NVIDIA Turing |
| Dedicated Memory Size(per GPU) | 11.81 GB |

### C.7.2  Training procedure

**Pretraining procedure:**  For CIFAR10 and CIFAR100, we train models using SGD with a momentum factor of 0.9 and weight decay of $5 \times 10^{-4}$, for 200 epochs using Cosine Annealing step sizes with an initial learning rate of 0.1.

**ImageNet post training:**  For ImageNet, we use off-the-shelf pretrained models from Torchvision [58]. We train the model for 3 epochs after each iteration of pruning with learning rates of 0.1, 0.01, 0.001. After the pruning ends, we finally train the network for 160 epochs with a batch size of 512. We use the SGD Optimizer with a momentum factor of 0.9 and weight decay of $1 \times 10^{-4}$ and start with an LR warm-up for 10 epochs, followed by Cosine Annealed step sizes with an initial learning rate of 0.1 with Cutmix and Mixup augmentations.

$L_2$ **Post training procedure:**  For the synthetic training experiments mentioned in Section 5, we first prune the model using $L_2$ norm as the grouped saliency to a similar sparsity as our algorithm. We then train the model using 50000 samples from the synthetic dataset for 100 epochs with a batch

size of 128 using SGD optimizer with momentum factor of 0.9 with initial learning rate of 0.01 and a MultiStepLR learning rate scheduler with milestones at 60 and 80 epochs.

# D   Additional Algorithm details

In this section, we discuss algorithmic nuances not discussed in the main body of the paper.

## D.1   Clarification on Lipschitz Bounds and Their Role

In Section 3, we introduced a local-to-global error bound (Theorem 3.6) that connects intermediate-layer deviations to changes in the final-layer output, assuming the model is composed of Lipschitz-continuous layers with constants $C_l$. This result serves to theoretically motivate the use of local reconstruction error – what we formalize as Subset Fidelity – as a proxy for reconstruction error at the output.

Importantly, we do not compute or estimate Lipschitz constants in any part of our algorithm. Our pruning and unlearning algorithms do not depend on knowledge of the values of $C_l$. The bound in Theorem 3.6 is used qualitatively to support the intuition that preserving high-fidelity intermediate representations leads to stability in the *final* model predictions.

Empirically, we find that Subset Fidelity correlates strongly with the effect of component removal on prediction quality (see Figure 1 and Appendix C), even in the absence of explicit Lipschitz bound estimation. This supports our design choice to treat Theorem 3.6 as a motivating principle, not an operational tool.

We believe this distinction is important to clarify: while our framework draws conceptual inspiration from Lipschitz continuity, it remains loss-free, and hyperparameter-driven in practice, with no reliance on any difficult-to-estimate constants.

## D.2   Additional Algorithmic details on Fidelity Estimation

To efficiently estimate the fidelity of each component at a given layer, we use a saliency measure to approximate the fidelity score. This is based on the component's contribution to reconstructing the layer's output, computed via the inner product between the layer output and the component-specific activation contribution. This can be written as

$$\tilde{R}_{ci}^l = \mathbb{E}\left[\langle \boldsymbol{Y}_c(\mathrm{X}), \boldsymbol{A}_{ci}(\mathrm{X})\rangle\right] = \langle \boldsymbol{Q}_i^c, \mathbf{1}\rangle = \alpha_{ci}^l \mathbb{E}[||\boldsymbol{A}_{ci}||^2]$$

In networks that include **BatchNorm** (e.g., ResNets [25], VGG [69]), we refine this reconstruction by centering the activations using the BatchNorm's stored running mean. This leads to a modified formulation of the component similarity matrix:

$$\tilde{Q}_{ij}^c = \mathbb{E}\left[\langle \boldsymbol{A}_{ci}(\mathrm{X}), \boldsymbol{A}_{cj}(\mathrm{X})\rangle\right] - \langle \mathbb{E}[\boldsymbol{A}_{ci}(\mathrm{X})], \mathbb{E}[\boldsymbol{A}_{cj}(\mathrm{X})]\rangle$$

These quantities are computed efficiently using modern GPU architectures. The forward activations from a calibration set are batched and evaluated across multiple GPUs in parallel. In sequence-based architectures such as Transformers [44], we compute the expectation over all elements in the sequence dimension.

**Numerical Stability and Regularization.**   To compute the optimal linear compensation for modifying components, we solve a least-squares system involving the component similarity matrix $\tilde{Q}^c$. However, this matrix may be ill-conditioned or rank-deficient in practice. To ensure numerical stability and avoid inversion errors, we add a small $\ell_2$ regularization term ($\lambda = 10^{-4}$) to the diagonal before solving.

**Behavior of HiFi Components During Editing.**   The role of HiFi components depends on the editing task:

- **Structured Pruning:** We retain HiFi components and discard the rest. While we cannot guarantee a fixed sparsity level in the output model (since HiFi components may span all inputs), we observe in practice that reasonable sparsity emerges naturally. For more aggressive pruning, the algorithm is applied iteratively.

- **Class Unlearning:** Simply discarding low-fidelity components is insufficient. Instead, we aim to remove or disrupt the influence of HiFi components that are specific to the forget class. The editing strategy depends on the network type:

  - In BatchNorm networks, we zero out the weights of HiFi components computed as per the forget class samples.
  - In LayerNorm-based networks with residual connections (e.g., Swin-T), we negate the weights of HiFi components. This rotates the forget-class representation in the opposite direction due to the residual path.

The unlearning strategy for Transformer-based architectures is captured in the following procedure:

---

**Algorithm 2** ViT-Edit-X: Structured Editing for Transformers

---

**Require:** Model parameters $\theta$, HiFi components $H$, coupled channels $CC$
**Ensure:** Edited model parameters $\hat{\theta}$
1: **if** $X = \texttt{Prune}$ **then**
2:    **for** $i \in [c_{\text{in}}^l] \setminus \{i \mid (c,i) \in \bigcup_l H_l\}$ **do**
3:       $\boldsymbol{W}_{c,i}^l \leftarrow 0$                            $\forall c \in [c_{\text{out}}^l],\ l \in CC$
4: **else if** $X = \texttt{Unlearn}$ **then**
5:    **for** each layer $l \in CC$ **do**
6:       $\hat{W}_{c,i}^l \leftarrow -W_{c,i}^l$                        $\forall (c,i) \in H_l$
7: **Return:** $\hat{\theta}$

---

**Fidelity Estimation in LLMs**   Due to the large scale of LLMs and the range of floating point values, estimation of scores becomes more challenging. We estimate the the fidelity scores by computing the row norms of the regularized Cholesky decomposition of $\boldsymbol{Q}$. The scores are estimated as

$$\text{FS}(\{i\}) \approx ||\boldsymbol{L}_i||^2 \quad \text{where} \quad \boldsymbol{Q} = \boldsymbol{L}\boldsymbol{L}^\top \quad \text{is the Cholesky decomposition of } \boldsymbol{Q}$$

We use the Cholesky decomposition since it is efficient to compute.

### D.3   Computational cost

Let $N$ be the number of data points used to estimate the saliency and $M^l$ be the complexity of computing the input contribution at layer $l$ for a single sample in a set of coupled channels with $m$ layers. The complexity to compute the set of retained channels for an output channel of a layer is, $t_{sal}^l = O(NM^lC_{in}^l d^l)$. To select the components for the coupled channels, the top $p$ elements for each layer and output channel in them are collected, this costs $O(\sum_{l=1}^m C_{out}^l(C_{in}^l \log C_{in}^l + t_{sal}^l))$. The algorithm shows a linear dependence on the number of layers in the network, compared with the BGSC algorithm [56] which has a quadratic dependence.

## E   Full LLM Disclosure

We employed Large Language Models (LLMs) to refine the text for grammar and clarity. Additionally, LLMs were used to generate auxiliary scripts for data visualization (plots). We confirm that LLMs were not used to implement any of the core algorithms or methodologies proposed in this work.

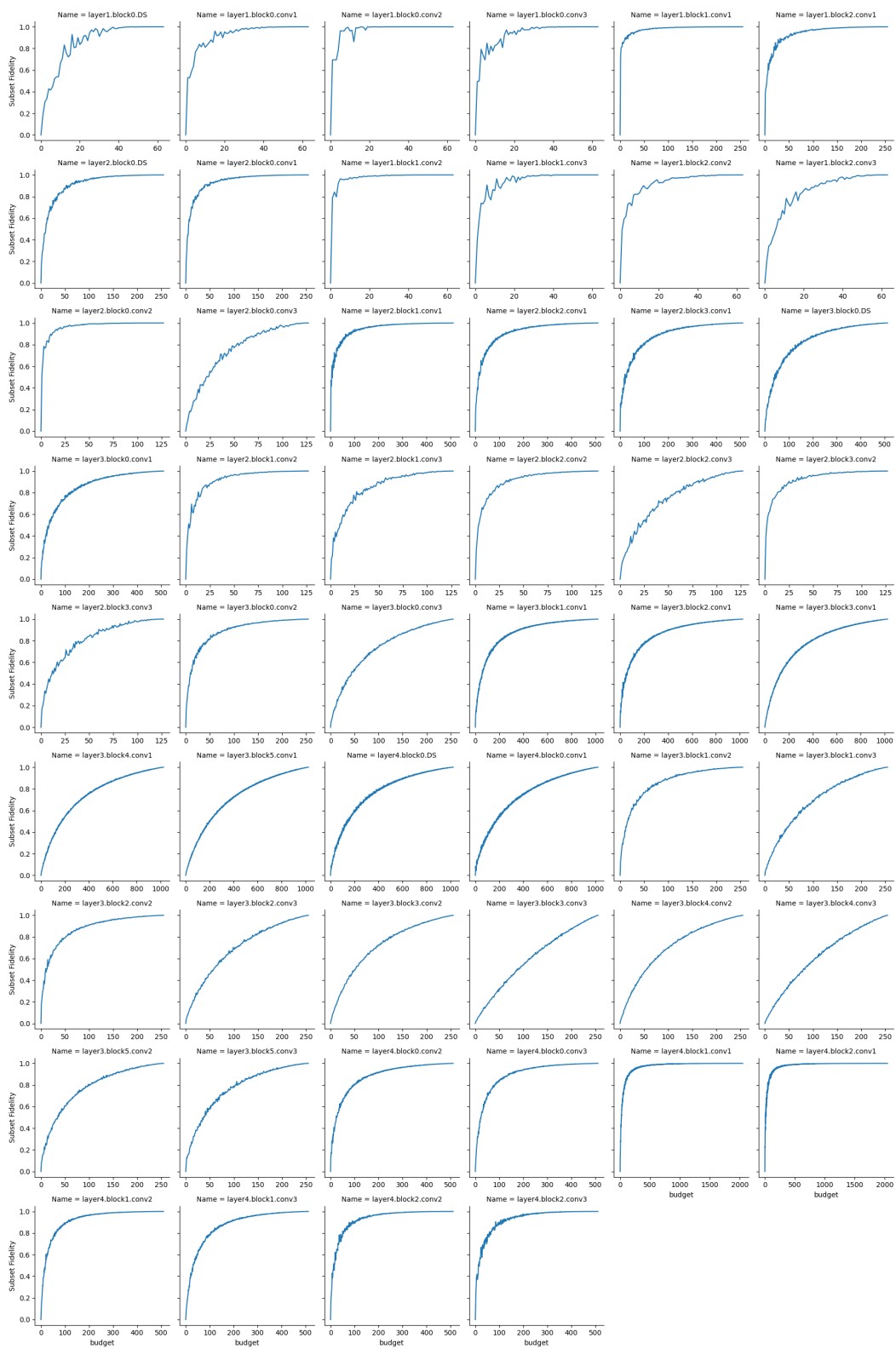

Figure 4: Estimates of Optimal subset fidelity for ResNet-50 on CIFAR10.

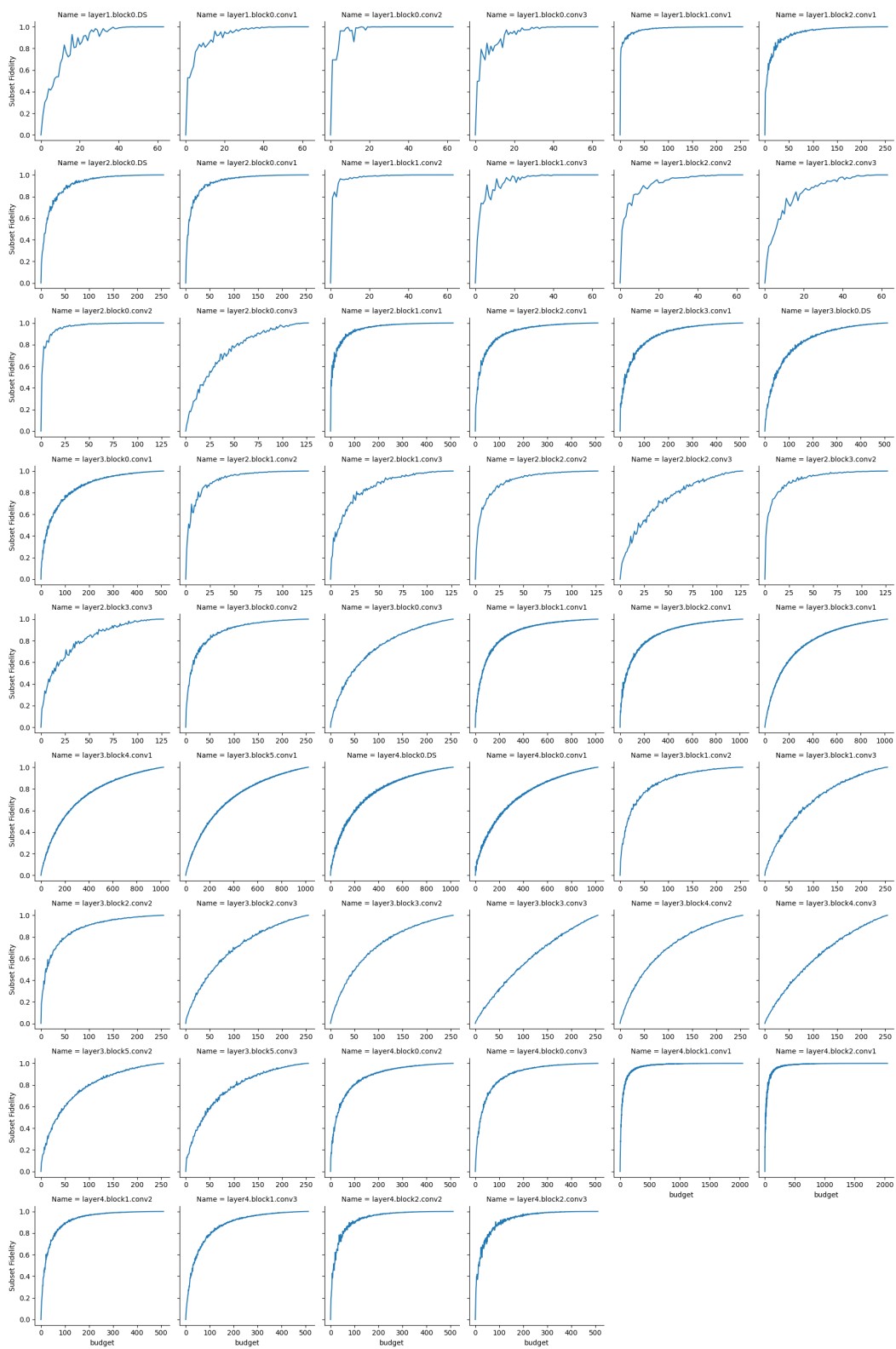

Figure 5: Estimates of Optimal subset fidelity for ResNet-50 on CIFAR100.

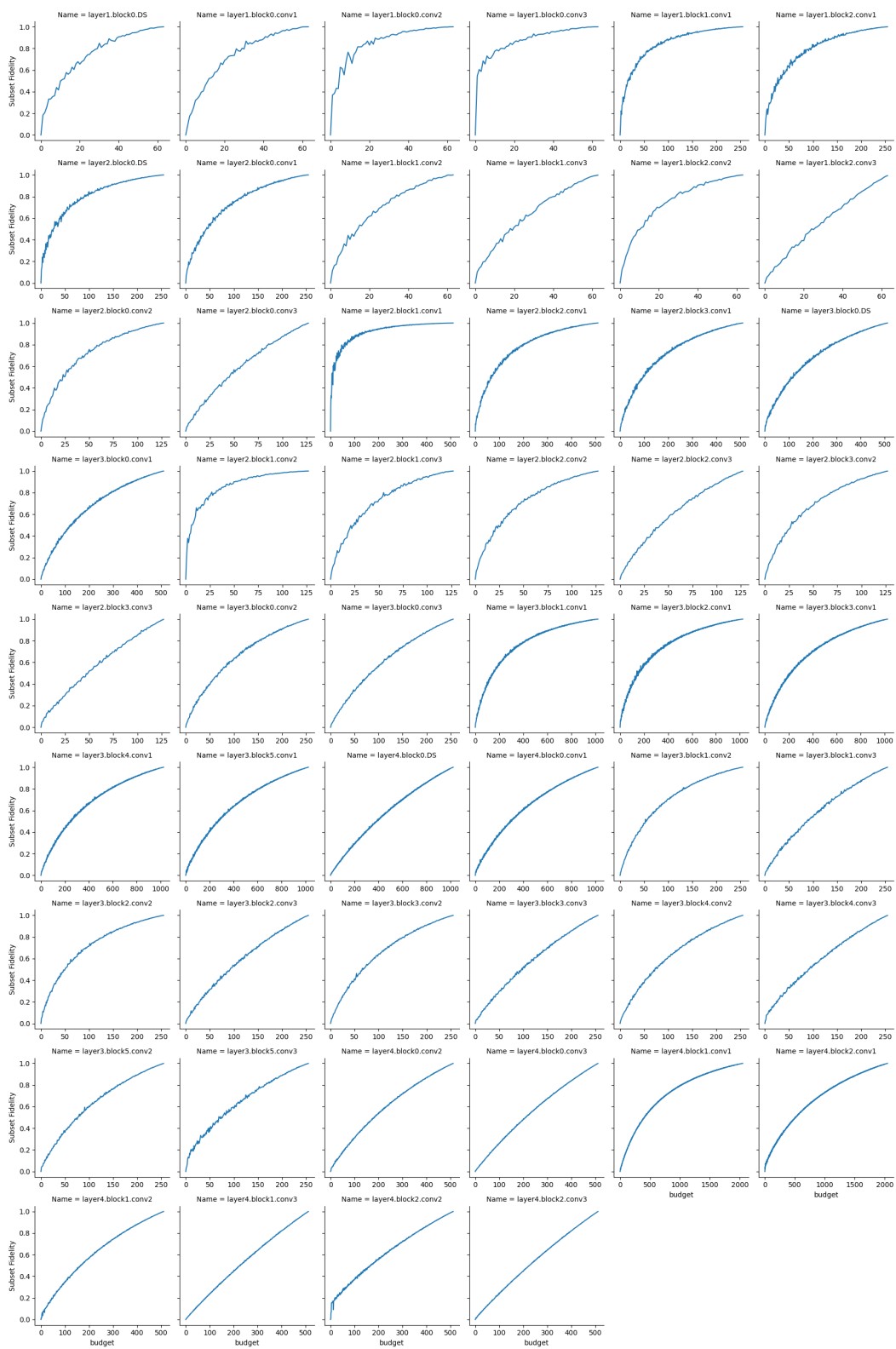

Figure 6: Estimates of Optimal subset fidelity for ResNet-50 on ImageNet.

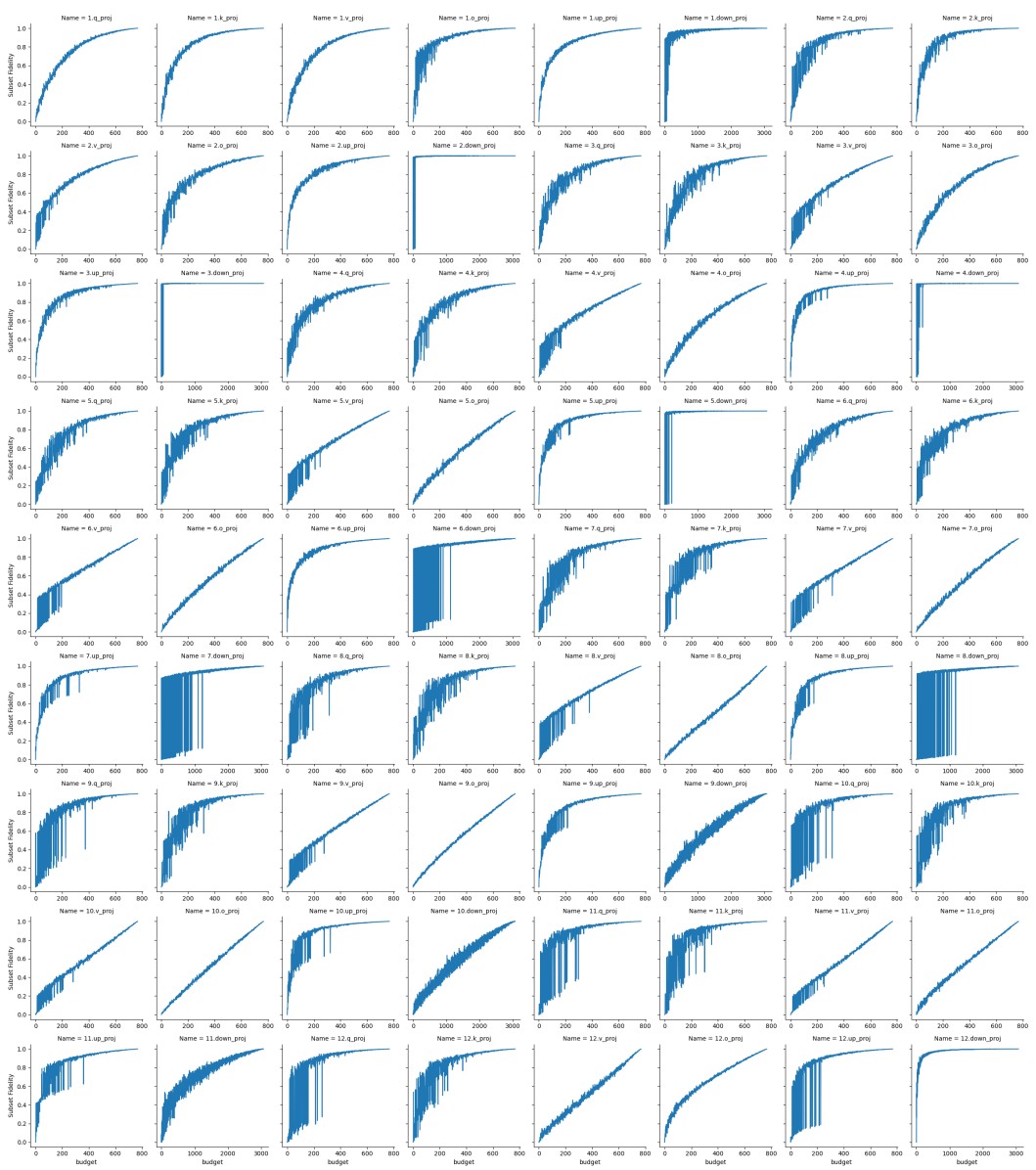

Figure 7: Estimates of Optimal subset fidelity for OPT-125M.

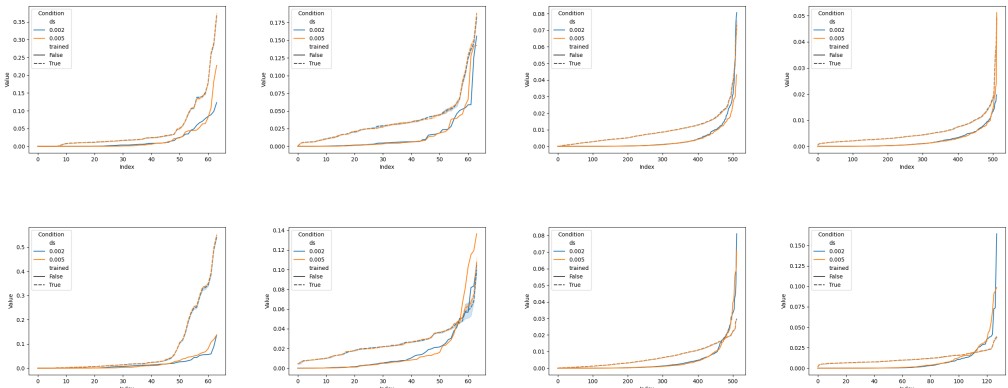

Figure 8: Fidelity scores for select layers of ResNet50 trained on ImageNet showing the effect of training and data set size (ds).

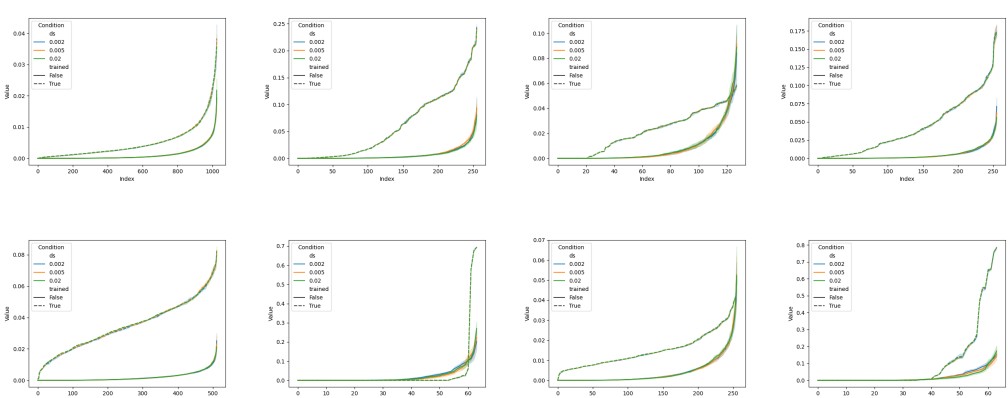

Figure 9: Fidelity scores for select layers of ResNet50 trained on CIFAR10 showing the effect of training and data set size.

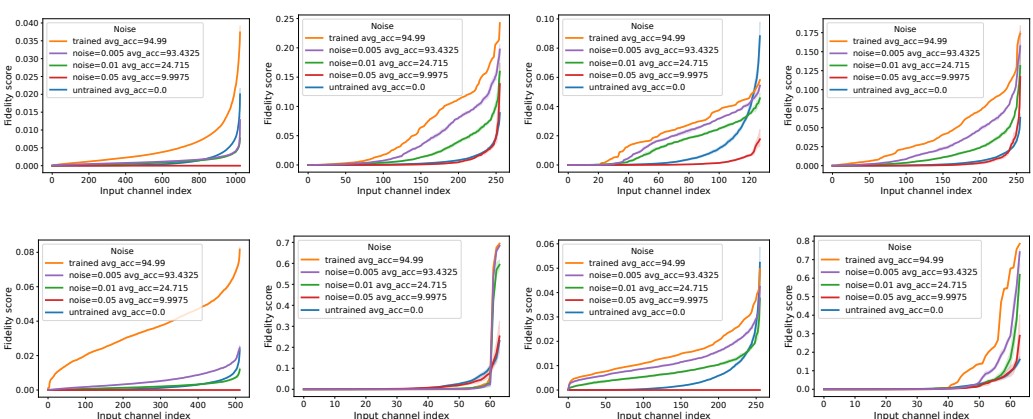

Figure 10: Fidelity scores for select layers of ResNet50 trained on CIFAR10 showing the effect of adding noise

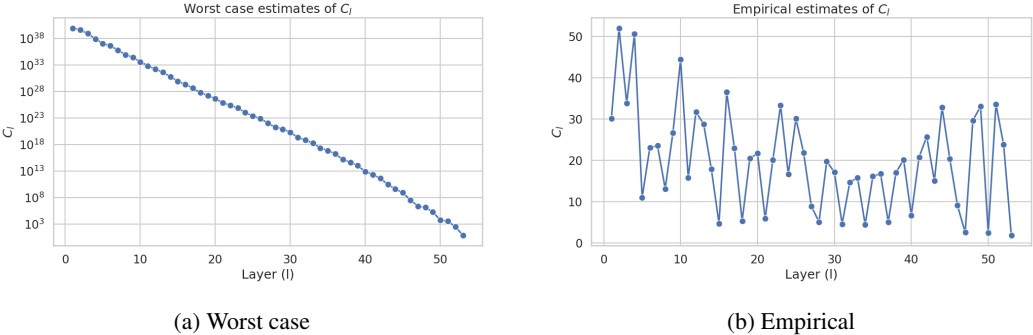

(a) Worst case                   (b) Empirical

Figure 11: Estimates of constants $C_l$ across layers for a ResNet-50 trained on ImageNet.

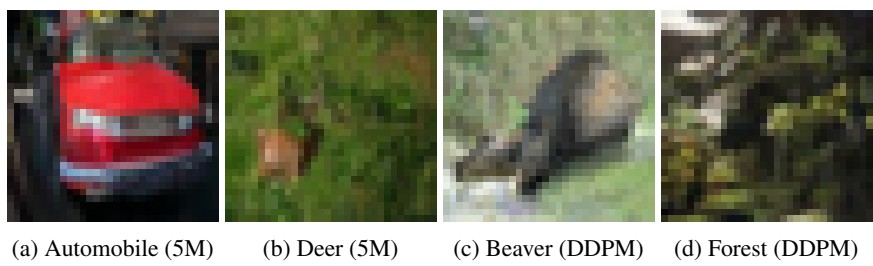

(a) Automobile (5M)     (b) Deer (5M)     (c) Beaver (DDPM)     (d) Forest (DDPM)

Figure 12: Randomly selected images from the synthetic sets

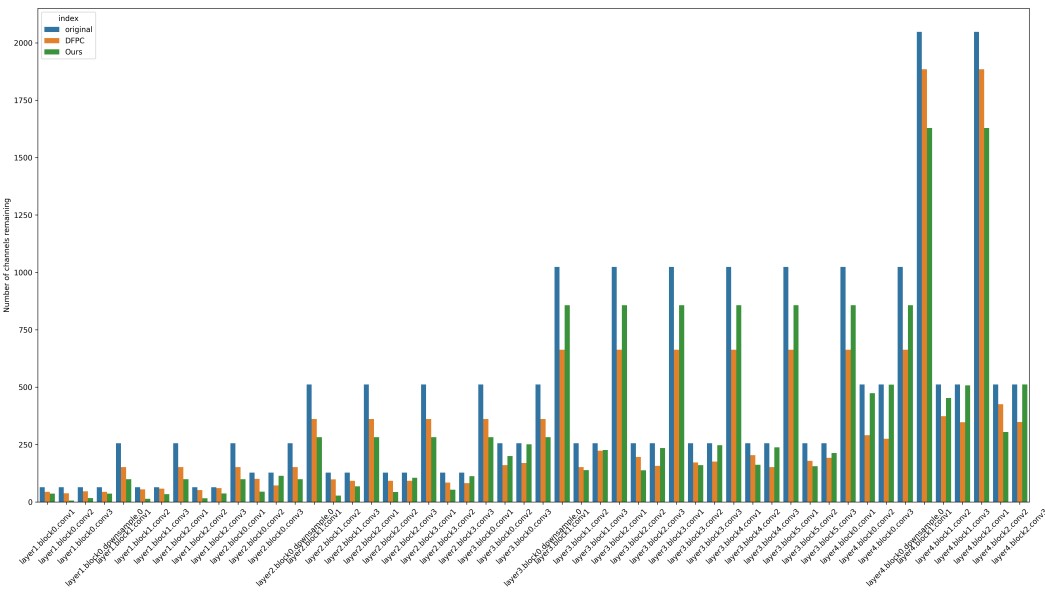

Figure 13: Number of remaining channels of pruned ImageNet model compared with DFPC (30)

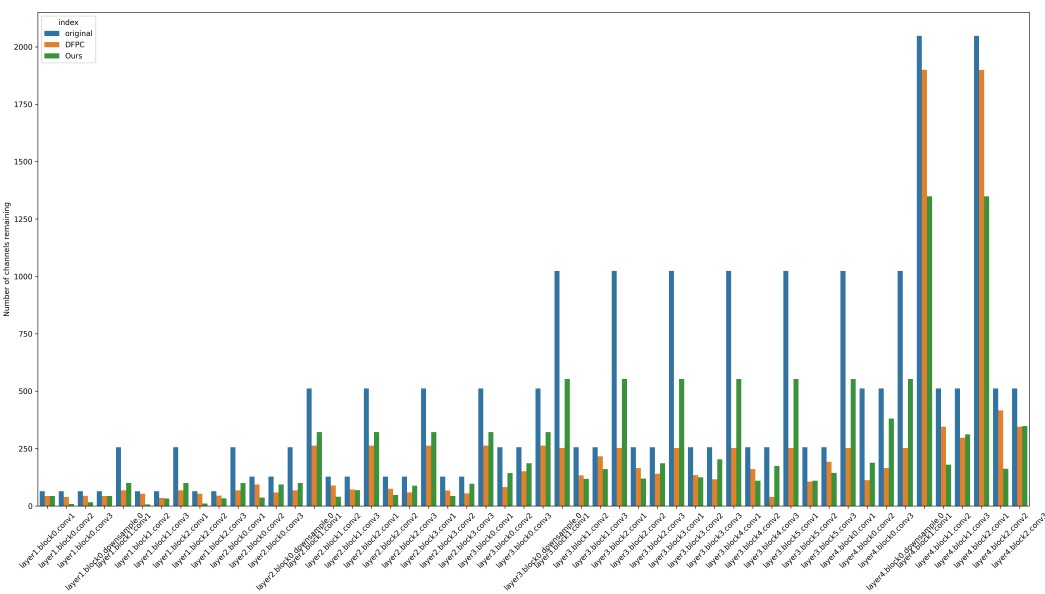

Figure 14: Number of remaining channels of pruned ImageNet model compared with DFPC (54)

