# OpenReview forum: "ModHiFi: Identifying High Fidelity predictive components for Model Modification"
_NeurIPS.cc/2025/Conference — NeurIPS 2025 spotlight_

### Official Review · Reviewer_17dD · 2025-06-22

**Clarity:** 3
**Significance:** 2
**Originality:** 2
**Rating:** 4
**Confidence:** 3

**Summary:**

This paper proposes ModHiFi, a pre-training data- and loss-free algorithm for model modification tasks such as pruning and unlearning. It introduces a theoretical method to identify critical model components and demonstrates competitive performance on both vision and language benchmarks.

**Questions:**

Q1: The term "model modification" may encompass diverse tasks beyond pruning/unlearning (e.g., model editing in LLM and some vision tasks in [2]). In addition, the paper exclusively evaluates zeroing out as the modification strategy. How might the identified components be adapted to other modification strategies (e.g., fine-tuning, structural reparameterization), and would their effectiveness generalize across different model modification tasks? Alternatively, it may be more appropriate to refine the terminology and scope description to better reflect the range of tasks actually validated in the experiments.

Q2. Under the same setting that uses synthetic image data for component identification in vision models, how does the proposed method compare to [2], which estimates component importances across the entire model rather than treating each layer separately?

Q3. In Table 3, the reported time cost for the proposed methods appears to exclude the Monte Carlo sampling-based identification step, which may make the comparison unfair.  Additionally, what is the time cost when applying the method to large language models?

Q4. Could the paper include comparisons with more recent methods, such as [3] (which appeared online before March 1st, 2025), to better position the proposed approach within the state-of-the-art?

[2] Decomposing and Editing Predictions by Modeling Model Computation, ICML 2024

[3] Numerical Pruning for Efficient Autoregressive Models, AAAI 2025

**Ethical Concerns:**

["NO or VERY MINOR ethics concerns only"]

**Final Justification:**

The rebuttal clarifies the Monte Carlo sampling and the model modification strategy, and discusses prior work. Therefore, I keep the current positive rating.

**Limitations:**

The method computes the fidelity of each layer independently, potentially overlooking inter-layer connections. Additionally, the computational cost of the Monte Carlo–based subset identification may become high when applied to large foundation models.

**Paper Formatting Concerns:**

Table 3 exceeds the text margin.

**Quality:**

3

**Strengths And Weaknesses:**

### Strengths
- The paper provides a theoretical analysis showing that, under Lipschitz continuity, the reconstruction error of the model is linearly bounded by the local reconstruction error.
- It introduces a Monte Carlo-based sampling strategy to identify subsets of important components using the proposed Subset Fidelity metric.
-The method is extensively validated on CNN and Transformer architectures, across vision and language tasks, demonstrating its generality and effectiveness.

### Weaknesses
-  Theorem 3.1 appears to be an extension or adaptation of prior work [1], which is not cited in the main paper.
- The method computes fidelity scores and prunes components independently for each layer, which potentially overlooks inter-layer dependencies among components during pruning.
- The paper would benefit from comparison with more recent baselines, where applicable, to ensure a fair and up-to-date evaluation.

[1] NISP: Pruning Networks using Neuron Importance Score Propagation, CVPR 2018

---

> ### Author Rebuttal · Authors · 2025-07-31
>
> We thank the reviewer for the positive evaluation of our work. We hope the additional clarifications provided here address the concerns and further demonstrate the significance and robustness of our work. We are eager to engage further with the reviewer during the discussion period to improve our work and scores in the process.
>
> We respond to their questions in the order that we believe addresses their most pressing concerns first.
>
> ## Questions
>
>
> **1. In Table 3, the reported time cost for the proposed methods appears to exclude the Monte Carlo sampling-based identification step, which may make the comparison unfair.**
>
> We would like to clarify to the reviewer that **Monte Carlo sampling of subsets is not a part of our pruning/unlearning algorithm**. The Monte Carlo experiment is an ablation study shown in Figure 1, as stated in line 203, to demonstrate the existence of small sets of high-fidelity components and provide context for Observation 1. Instead, as mentioned in lines 201-202, we approximate the solution by constructing the subset of a required size with the largest individual fidelity scores. We further clarify that we perform random sampling of a set of synthetic data to estimate the statistics of the intermediate solution. The experiment described in Figure 1 adds empirical validation to our observations.
>
> **2. [Continuation to your question on Monte-Carlo Simulation experiments] Additionally, what is the time cost when applying the method to large language models?**
>
> To perform the Monte Carlo Simulations on LLMs is expensive. It took us up to 20 hours to conduct this experiment on the OPT-125m, as shown in Figure 1 on the hardware described in Appendix C.5. However, we reiterate that this Monte-Carlo step is not a part of our model-modification algorithm and is only used to demonstrate the existence of small sets of high-fidelity components in various architectures.
>
>
> **3. The paper exclusively evaluates zeroing out as the modification strategy. How might the identified components be adapted to other modification strategies (e.g., fine-tuning, structural reparameterization), and would their effectiveness generalize across different model modification tasks?**
>
> In this work, we focus on the identification of components important for predictions without access to gradients or loss functions. We choose pruning and unlearning as they are two tasks that directly benefit from identification in this setting. We hope the strong empirical results from this work encourage further interest in the community in applying these techniques to additional tasks. We would like to point out, per our definition of model modification as stated in Equation MODIFY (line 109), structural reparametrization is not in our scope of work. Moreover, for finetuning, we precisely aim to eliminate it to reduce the computational burden for model modification, which is the goal of this work.
>
> **4. The term "model modification" may encompass diverse tasks beyond pruning/unlearning (e.g., model editing in LLM and some vision tasks in [2])...Alternatively, it may be more appropriate to refine the terminology and scope description to better reflect the range of tasks validated in the experiments.**
>
> We thank the reviewer for the suggestion and will refine the terminology to contextualize the experimental evaluation.
>
> **5. Under the same setting that uses synthetic image data for component identification in vision models, how does the proposed method compare to [2], which estimates component importances across the entire model rather than treating each layer separately?**
>
> We would like to note that [2] requires access to the loss function and access to training data to compute a brute force estimate of component importance across the entire network. In our work, we assume the setting wherein the loss function is not available, and we propose a technique that does not require access to either of these. Moreover, their work is not architecture-agnostic. Our results indicate that even treating each layer separately is sufficient to provide strong performance on pruning and unlearning tasks. This assumption has become standard in literature due to the large size of the models [4, 5, 6,7, 8, 9]. However, equipped with Theorem 3.1, we can bound the error in the last layer linearly in terms of the local error, indicating that for such networks, the propagation of local error across layers is not severe, and can be upper-bounded by a linear function of the local error in the worst case.
>
> **6. Could the paper include comparisons with more recent methods, such as [3] (which appeared online before March 1st, 2025), to better position the proposed approach within the state-of-the-art?**
>
> We thank the reviewer for drawing our attention to this work. We will include this in our related work and provide context for recent pruning methods. As for direct comparisons, [3] does not report results on the models considered in this work and does not make its code available. In terms of differences in pruning methodology, they propose an iterative scheme for computing pruning-specific scores, while we propose a single-shot scheme for computing general-purpose importance scores that can be applied to a variety of tasks. Additionally, it is not clear how this would apply to other modification tasks or convolution architectures.
>
> ---
>
>
> ## Summary and Strength
>
> We thank the reviewer for appreciating the theoretical analysis and extensive empirical validation of our work. While we appreciate the summary and the strengths, we would like to bring to the reviewer’s notice a clarification. Our proposed method is also gradient-free and does not require computing gradient-based updates to identify components or compensate for lost statistical performance.
>
> ---
>
>
> ## Weaknesses and Limitations
>
> **Lack of mention of related work**
>
> We thank the reviewer for bringing [1] to our notice. We will reference this work in the paper and highlight how our work differs from it in our related work section. In this work, we specifically consider the statistics of the intermediate distributions to inform our algorithm by upper-bounding the expected error, whereas [1] considers assigning importance scores without taking the distribution of data into account, providing a more conservative bound. Moreover, their theorem only holds for feed-forward layers with Lipschitz non-linear functions and doesn’t generalize to other architectures such as transformers. Our work also addresses networks with complex interconnections. Additionally, [1] focuses only on pruning, whereas this work focuses on the identification of components essential for prediction.
>
> **Expensive Monte Carlo Sampling**
>
> We refer the reviewer to our response to Question 1.
>
> **Inter-layer dependency**
>
> We refer the reviewer to our response to Question 5.
>
>
> **We thank the reviewer for their thorough review, which has enabled us to clarify the key claims of our work. In particular, the reviewer helped us contextualize the Monte Carlo experiments showing the existence of small subsets of high-fidelity components. We hope this discussion results in an increase in our scores, and we are eager for further discussion on the merits of our work.**
>
> ## References
>
> [1] NISP: Pruning Networks using Neuron Importance Score Propagation, CVPR 2018
>
> [2] Decomposing and Editing Predictions by Modeling Model Computation, ICML 2024
>
> [3] Numerical Pruning for Efficient Autoregressive Models, AAAI 2025
>
> [4] Elias Frantar, Sidak Pal Singh, and Dan Alistarh. Optimal brain compression: A framework for accurate post-training quantization and pruning, 2022.
>
> [5] Hao Li, Asim Kadav, Igor Durdanovic, Hanan Samet, and Hans Peter Graf. Pruning filters for efficient convnets. In International Conference on Learning Representations, 2017.
>
> [6] Jian-Hao Luo, Jianxin Wu, and Weiyao Lin. Thinet: A filter level pruning method for deep neural network compression. In Proceedings of the IEEE International Conference on Computer Vision, pages 5058–5066, 2017.430
>
> [7] Xinyin Ma, Gongfan Fang, and Xinchao Wang. Llm-pruner: On the structural pruning of large language models. Advances in neural information processing systems, 36:21702–21720, 2023.
>
> [8] Xin Men, Mingyu Xu, Qingyu Zhang, Bingning Wang, Hongyu Lin, Yaojie Lu, Xianpei Han, and Weipeng Chen. Shortgpt: Layers in large language models are more redundant than you expect. arXiv preprint arXiv:2403.03853, 2024.
>
> [9] Elias Frantar and Dan Alistarh. SparseGPT: Massive Language Models Can Be Accurately Pruned in One-Shot, 2023.

---

> > ### Comment · Reviewer_17dD · 2025-08-05
> >
> > Thanks for the detailed rebuttal. I appreciate the clarifications on Monte Carlo sampling and the model modification strategy, and I acknowledge your plans to refine terminology and update related work. I have no further questions and will keep my current rating.

---

> ### Author Response · Authors · 2025-08-07
>
> Dear reviewer 17dD,
>
> As indicated by your previous response, we are happy that our rebuttal was able to address your concerns satisfactorily. In response to the concerns raised, we’ve made several changes that have strengthened the manuscript, especially,
>
> - We have provided additional clarification on the Monte Carlo experiment presented in our work, which has helped improve the clarity of our manuscript
> - Contextualize our theoretical results with prior work, specifically showing how our work is more than an extension of NISP [1]
>
> In light of the extension of the discussion period, and in the interest of further strengthening our work, we would be pleased to learn if there are any suggestions within the scope of the paper that would prompt the reviewer to increase their rating.

---

### Official Review · Reviewer_Ze6v · 2025-07-01

**Clarity:** 2
**Significance:** 3
**Originality:** 2
**Rating:** 4
**Confidence:** 3

**Summary:**

This paper presents ModHiFi, a layer-wise reconstruction-based method for estimating feature importance in neural networks using synthetic data, avoiding the need for original training data. It introduces the High Fidelity (HiFi) metric, which quantifies the contribution of each component to local reconstruction. This importance score is used for both structured pruning (retaining high-HiFi components) and classwise unlearning (removing high-HiFi components). The method leverages the empirical observation that pretrained Transformers exhibit approximate Lipschitz continuity, and uses this to theoretically justify that local reconstruction errors can linearly bound global prediction degradation. ModHiFi is evaluated on CNNs and Transformers across CIFAR-10 and ImageNet, showing competitive performance without access to the original training data.

**Questions:**

1. **Mitigating theoretical limitations in simultaneous pruning**: Theorem 3.1 assumes that inputs to each layer remain fixed, yet ModHiFi prunes multiple layers simultaneously. Could you clarify how the theoretical guarantees could hold under this setup? It might be worth considering including an empirical comparison between layerwise iterative and simultaneous pruning to assess the bound’s practical validity.

2. **Addressing data sensitivity more explicitly**: The reliance on synthetic or calibration data is mentioned in the appendix but is central to the method’s performance. The paper would benefit from moving this discussion into the main text and expanding it with quantitative evaluations showing how fidelity scores and performance vary with data size, diversity, or distribution mismatch / dataset sfhit.

3. **Quantifying computational overhead**: ModHiFi requires activation logging and per-layer evaluation for each candidate component. It could be worth providing runtime, memory, or efficiency comparisons with standard pruning baselines (e.g., L1-norm). Furthermore frontier plots of sparsity-accuracy tradeoffs would be helpful to position ModHiFi against related work.

**Ethical Concerns:**

["NO or VERY MINOR ethics concerns only"]

**Final Justification:**

The strong rebuttal clarified and addressed my mentioned concerns.

**Limitations:**

The authors acknowledge certain limitations, such as the method’s reliance on pretrained models and the intractability of full subset selection, which are briefly discussed in the main text. However, additional limitations - particularly the method’s dependence on data quality for synthetic inputs should be more explicitly addressed, and ideally consolidated.

**Paper Formatting Concerns:**

No concerns

**Quality:**

3

**Strengths And Weaknesses:**

Strengths
- **Works for both pruning and unlearning**: ModHiFi applies the same reconstruction-based importance score to both structured pruning and classwise unlearning, demonstrating that the metric captures feature relevance across different model editing tasks.
- No reliance on training data: The method does not require access to the original training set or supervised loss function. Instead, it operates using either synthetic inputs or a calibration set, making it significant for pre-trained models and their applications.
- **Theoretical support via Lipschitz continuity**: By leveraging the Lipschitz property of trained networks, the paper provides a theoretical guarantee that local reconstruction errors upper-bound global output degradation. This justifies the use of local fidelity scores for importance estimation, and makes the approach more practical by avoiding the need for end-to-end evaluations.
- **Empirical validation of the importance metric**: The paper includes several supporting experiments and ablations, such as noise injection that empirically demonstrate the HiFi score correlates with true functional importance.
- **Empirical effectiveness**: Despite operating without training data, gradients, or loss access, the method achieves competitive performance across pruning and unlearning on CIFAR-10 and ImageNet.

Weaknesses
- **Limited novelty and insufficient comparison to prior reconstruction-based pruning methods**: While ModHiFi is framed as novel, the core idea of using reconstruction error to guide pruning, has been well explored in prior literature [1,2,3,4]. The paper does not adequately position its contributions relative to this body of work or clarify what distinguishes ModHiFi’s reconstruction-based formulation beyond its application to multiple tasks. Moreover, the practical implementation relies on singleton importance scores due to the infeasibility of true subset selection, which diminishes the claimed departure from per-component heuristics.

- **Dependence on synthetic or calibration data**: While ModHiFi avoids training data, it still relies on synthetic or curated datasets to compute fidelity scores and fit compensation terms. The quality, representativeness, and diversity of this data can significantly impact performance, yet the main paper offers limited discussion or quantification of this sensitivity. Additionally, fitting compensation weights via linear regression constitutes a lightweight form of fine-tuning, which somewhat undermines the “no fine-tuning” framing.

- **Theoretical guarantees rely on unrealistic assumptions**: Theorems 3.1 and 3.7 rely on assumptions that rarely hold in practice. The uncorrelated feature assumption underlying singleton optimality is implausible in deep networks, where strong interdependencies exist. Similarly, Theorem 3.1 assumes that each layer’s inputs remain fixed during analysis, yet ModHiFi prunes multiple layers simultaneously, violating this condition.

- **Lack of computational cost analysis**: Although the method is presented as more efficient than fine-tuning, the paper does not quantify its computational demands. Evaluating fidelity scores requires full forward passes and activation logging across a dataset for each candidate component, which may scale poorly in large networks. The absence of runtime comparisons or memory profiling against baseline methods (e.g., magnitude-based pruning) leaves open questions about ModHiFi’s practical efficiency.


[1] Jiang, Chunhui, et al. Efficient DNN Neuron Pruning by Minimizing Layer-wise Nonlinear Reconstruction Error. IJCAI, 2018.
[2] Kamma, Koji, and Toshikazu Wada. Reconstruction Error Aware Pruning for Accelerating Neural Networks. ISVC, 2019.
[3] Zhuang, Zhuangwei, et al. Discrimination-aware Channel Pruning for Deep Neural Networks. NeurIPS, 2018.
[4] Molchanov, Pavlo, et al. Importance Estimation for Neural Network Pruning. CVPR, 2019.

---

> ### Author Rebuttal · Authors · 2025-07-31
>
> We thank the reviewer for providing a comprehensive review of our work. We are grateful that you find value in our theory and empirical performance, and wish to provide clarification to the questions raised by the reviewer.
>
> ## Summary and Strengths
>
> We are happy to note that the reviewer finds several strengths in our work. Specifically, its broad applicability, theoretical support, and empirical effectiveness. However, we would like to highlight some points in our work that the reviewer might have missed, particularly that **we also show the broad applicability of our results to LLMs.**
>
> ---
> ## Questions and Weaknesses
>
>
> **1. Mitigating theoretical limitations in simultaneous pruning: Theorem 3.1 assumes that inputs to each layer remain fixed, yet ModHiFi prunes multiple layers simultaneously. Could you clarify how the theoretical guarantees could hold under this setup?**
>
> We would like to clarify that **Theorem 3.1 does not assume inputs are fixed**, only the weights of layers after the layer being investigated are fixed (say layer $l$). All layers before layer $l$ can be modified, and the reconstruction error at the output would still be bounded by that of layer $l$.
>
> We also want to clarify that **Algorithm 1 on page 6 is applied to a single layer**. In practice, we find that the best performance is achieved when layers are modified iteratively, starting from the initial layer for pruning. For unlearning, we find we only need to iteratively modify the last few layers, as specified in Appendix C.4.
>
>
> **2. Addressing data sensitivity more explicitly/Dependence on synthetic or calibration data: The reliance on synthetic or calibration data is mentioned in the appendix, but is central to the method’s performance.**
>
> We first clarify that **our method itself does not require synthetic data**. Rather, we address the problems of pruning and unlearning in the restrictive setting where the training data and loss function used to train the model are unavailable, but with distributional access. Our method addresses that problem and is effective even when only synthetic data can be used.
>
> **3. The paper would benefit from moving this discussion into the main text…the main paper offers limited discussion or quantification of this sensitivity.**
>
> As we have **already discussed in Appendix C.2**, we show that **higher quality synthetic data leads to better estimation and identification of high-fidelity components and performance for pruning**, as illustrated in Table 4 of the Appendix. Furthermore, as we show in [our response to reviewer qfdv](https://openreview.net/forum?id=lClK4uBxSG&noteId=4eCrWRgDBD), **using the original training data for identifying HiFi components for unlearning tasks results in superior unlearning performance than with synthetic data.** We have also provided **experiments highlighting the efficacy of fine-tuning a pruned model (using $L_2$ norms of weights) with synthetic data instead of the original training set in Appendix C.3, Table 5**, as well as Table 1 of the main document. We show that low-quality synthetic data is insufficient for fine-tuning heavily pruned models. If accepted, we will move this discussion from the Appendix to the main document.
>
> **4. Quantifying computational overhead/ Lack of computational cost analysis… Although the method is presented as more efficient than fine-tuning, the paper does not quantify its computational demands...absence of runtime comparisons or memory profiling…**
>
> We kindly refer the reviewer to Appendix C.5, where we have **already discussed the theoretical and practical computational cost of the ModHiFi algorithm.** In particular, we showcase that **fidelity scores can be computed in roughly 5 minutes for ResNet-50 models trained on Imagenet**, and **compensation terms can be computed in roughly 1 minute per layer on average.** We also point out that while using weight norms for pruning would undoubtedly be faster than ModHiFi, purely weight-based techniques cannot be broadly applied to other tasks, such as unlearning. Additionally, norm-based techniques are not as performant in retaining accuracy, as noted in Table 1.
>
> **5.  Furthermore frontier plots of sparsity-accuracy tradeoffs would be helpful to position ModHiFi against related work.**
>
> We will provide such plots in our revised manuscript. However, we regretfully remind the reviewer that **we are prohibited from providing any plots during the rebuttal period**.
>
> **6. Limited novelty and insufficient comparison to prior reconstruction-based pruning methods: While ModHiFi is framed as novel, the core idea of using reconstruction error to guide pruning, has been well explored in prior literature [1,2,3,4]. The paper does not adequately position its contributions relative to this body of work or clarify what distinguishes ModHiFi’s reconstruction-based formulation beyond its application to multiple tasks.**
>
> We politely disagree that the novelty of our work is limited. As stated by reviewer qfdv, the applicability of our approach to a variety of tasks is a strength of our work and is novel. Specifically, the applicability of **reconstruction error to Unlearning has not been addressed in prior art.** Moreover, unlike prior art, our proposed method identifies **groups of components in connected layers** (such as layers connected by residual connections in ResNet and transformer models) **as noted in Section 4 and Appendix D.2**, which has not been addressed in prior art. Moreover, reconstruction error is a metric used to form a basis for our model modification algorithms, which are novel and have novel consequences, as also noted by other reviewers.
>
> **7. Moreover, the practical implementation relies on singleton importance scores due to the infeasibility of true subset selection, which diminishes the claimed departure from per-component heuristics.**
>
> Our work provides singleton importance scores as a heuristic for solving an otherwise NP-complete problem, which can then be applied to a variety of model modification tasks, such as pruning and unlearning
>
> **8. Additionally, fitting compensation weights via linear regression constitutes a lightweight form of fine-tuning, which somewhat undermines the “no fine-tuning” framing.**
>
> Following recent work such as [5,6], fitting compensation weights does *not* constitute lightweight fine-tuning and is considered to be part of the pruning methods in general. However, if the reviewer insists, we will modify the manuscript to make this distinction by using the term “gradient-based fine-tuning.”
>
> **9. Theoretical guarantees rely on unrealistic assumptions…**
>
> Theorem 3.7 serves to highlight that there are cases when the greedy pruning strategy is optimal, even if it is not the case generally, **motivating its use as a useful heuristic** for solving an NP-hard problem. For a practical setting, we rely on our empirical results to justify the merit of our model modification algorithms. As already mentioned in Appendix C.1.2 and Appendix C.1.3, we conduct several experiments to show that the subsets generated by our algorithm are correlated to the final prediction by showing that these components are disproportionately responsible for model’s prediction.
>
> **10. Similarly, Theorem 3.1 assumes that each layer’s inputs remain fixed during analysis, yet ModHiFi prunes multiple layers simultaneously, violating this condition.**
>
> The purpose of Theorem 3.1 is to highlight the fact that the error in the output layer is at worst linear in the error at any intermediate layer, and serves to provide a principled grounding for using local reconstruction for identifying important components for pruning and unlearning.
>
> ---
> **We thank the reviewer for their comprehensive review, which has helped us improve our submission. We would like to invite them to engage with the discussion period to discuss and clarify any other concerns preventing them from increasing their score and accepting this work.**
>
> ---
>
> ## References
>
> [1] Jiang, Chunhui, et al. Efficient DNN Neuron Pruning by Minimizing Layer-wise Nonlinear Reconstruction Error. IJCAI, 2018.
>
> [2] Kamma, Koji, and Toshikazu Wada. Reconstruction Error Aware Pruning for Accelerating Neural Networks. ISVC, 2019.
>
> [3] Zhuang, Zhuangwei, et al. Discrimination-aware Channel Pruning for Deep Neural Networks. NeurIPS, 2018.
>
> [4] Molchanov, Pavlo, et al. Importance Estimation for Neural Network Pruning. CVPR, 2019.
>
> [5] Shah et al. Decomposing and Editing Model Predictions by Modeling Model Computations. ICML 2024.
>
> [6] Hoefler et al. Sparsity in Deep Learning: Pruning and growth for efficient inference and training in neural networks. JMLR, 2021.

---

> > ### Comment · Area_Chair_cc2X · 2025-08-04
> >
> > Dear Reviewer,
> >
> > Many thanks for your very structural comments. Please confirm if you have read the response. Can you take a look at the response and check if they solved your concerns or have followup questions.
> >
> > Best,
> > AC

---

> > ### Comment · Reviewer_Ze6v · 2025-08-04
> >
> > I appreciate the detailed rebuttal and am satisfied with the response.

---

> > > ### Author Response · Authors · 2025-08-04
> > > **Thank you for your response to our rebuttal**
> > >
> > > Dear Reviewer Ze6v
> > >
> > > Thank you very much for responding to our rebuttal. We are glad that we have been able to address your concerns.

---

> ### Author Response · Authors · 2025-08-07
>
> Dear reviewer Ze6v,
>
> As indicated by your previous response, we are happy that our rebuttal was able to address your concerns satisfactorily.
> We are especially pleased that our clarifications have sufficiently answered your concerns and you have chosen to increase your score.
> In response to the concerns raised, we’ve made several changes that have strengthened our work. In particular, we would like to thank you for helping to
>
> - Significantly improve the discussion on the dependence of data quality and clarify the notion of distributional access
> - Provide additional discussion on the computational cost of our proposed method
>
> In light of the extension of the discussion period, and in the interest of further strengthening our work, we would be pleased to learn if there are any suggestions within the scope of the paper that would prompt the reviewer to further increase their rating.

---

### Official Review · Reviewer_qfdv · 2025-07-02

**Clarity:** 3
**Significance:** 2
**Originality:** 3
**Rating:** 4
**Confidence:** 3

**Summary:**

The paper introduces ModHiFi, a novel framework for modifying well-trained models without access to the original training data or loss function. The authors address two key tasks: structured pruning and classwise unlearning. The core idea revolves around identifying High-Fidelity (HiFi) components which are small subsets of model components critical to predictive performance, using a new metric called Subset Fidelity, which quantifies how well a subset of components can reconstruct model outputs after weight adjustments.

**Questions:**

See concerns above.

**Ethical Concerns:**

["NO or VERY MINOR ethics concerns only"]

**Final Justification:**

The rebuttal addressed my questions. But after checking other reviews, I feel the paper is not good enough for an accept. I will keep my rating.

**Limitations:**

Yes

**Quality:**

3

**Strengths And Weaknesses:**

**Strength:**
+ Architecture Generalization: The proposed algorithm performs well on both vision models (CNNs/ViTs) and language models (LLMs), demonstrating versatility across different architectures.
+ Task Generalization: The modification based on the HiFi objective unifies two tasks, classwise unlearning and structured pruning. Specifically: The pruning method (ModHiFi-P) achieves an 11% speedup on ImageNet while maintaining superior accuracy over baselines. The unlearning method (ModHiFi-U) completely erases target classes without fine-tuning and is 10x faster than existing approaches.
+ Comprehensive Experimental Setup: The experiments thoroughly address the four key questions posed in the paper.

**Weakness:**
- As shown in Table 2, while the method maintains strong average performance under high pruning ratios, its individual task metrics are not particularly outstanding.
- The method’s effectiveness relies on the assumption that the target model is well-trained; its performance on poorly initialized or underfitted models remains unexplored and is theoretically expected to degrade.
- The classwise unlearning performance on Swin-Transformers is subpar, possibly due to insufficient model training.
- Computing Subset Fidelity requires high-dimensional matrix inversion, which may be computationally prohibitive for extremely large models (e.g., LLMs).
- The experimental scope is somewhat limited:
1) No exploration of the impact of k and η (key hyperparameters in HiFi selection).
2) No discussion on the choice of B (pruning budget) for structured pruning or the sizes of forget set and retain set for classwise unlearning.

---

> ### Author Rebuttal · Authors · 2025-07-31
>
> We thank the reviewer for the positive appraisal of our work, in particular, your note about the generalisability of our work to different architectures and model modification tasks.  We hope the additional results and clarifications provided here address the concerns and further demonstrate the significance and robustness of our work. We are eager to engage further with the reviewer during the discussion period regarding our work and, in the process, improve our scores too.
>
> ## Response to Questions and Weaknesses
>
>
> **1. As shown in Table 2, while the method maintains strong average performance under high pruning ratios, its individual task metrics are not particularly outstanding.**
>
> We would like to gently point out that on lower sparsity ratios (10%, 20%), our approach achieves the best performance on 80% of tasks, and is better than or competitive with the nearest baselines at higher sparsity as well. The variable performance on specific tasks can be seen across the baselines as well as shown in Table 2 of our main paper.
>
> **2. The method’s effectiveness relies on the assumption that the target model is well-trained; its performance on poorly initialized or underfitted models remains unexplored and is theoretically expected to degrade.**
>
> We gently point out that only well-trained models are relevant to this work, as common to the structured pruning literature for both vision and language models, under-trained/underfitted models are not considered for pruning [1, 2]. Moreover, unlearning cannot be defined for a poorly trained model, as such models haven’t learned sufficiently to make unlearning meaningful.
>
> **3. The classwise unlearning performance on Swin-Transformers is subpar, possibly due to insufficient model training.**
>
> We point out that the unlearning performance results for Swin-Transformers presented in the main paper are **in the more restricted setting without any fine-tuning and using synthetic data only**, whereas the nearest baselines all perform 10 epochs of fine-tuning using the original training data. Moreover, we show in Appendix C that performing only 3 epochs of fine-tuning achieves just 0.1% accuracy on the forget class.
>
> We also present an experiment comparing the efficacy of our unlearning methods with 10 epochs of fine-tuning, matching those of the baseline [3], both with synthetic data as well as the training data in the table below. This experiment highlights the efficacy of our unlearning method by **demonstrating *perfect unlearning* when using the CIFAR10 train set and superior remain class accuracy as well, outperforming the baseline from [3] when the same extensive retraining and the training set are available to us.**
>
> ---
> | Approach | Dataset | Forget Acc. | Remain Acc. |
> |--|--|--|--|
> | Jia et al | CIFAR10 Train | 1.20% | 90.69% |
> | Ours | CIFAR10 Synthetic |0 .37% | 84.63% |
> |Ours | CIFAR10 Train | 0.00% | 91.1% |
> ---
>
> **4. Computing Subset Fidelity requires high-dimensional matrix inversion, which may be computationally prohibitive for extremely large models (e.g., LLMs).**
>
> We clarify that **no actual matrix inversion is required in our work**. The least squares problems are solved by solving linear equations. For language models, we use a Cholesky decomposition of the $Q$ matrices to further improve numerical performance, as discussed in Appendix D. As we have already discussed in Appendix C.5.1, our approach allows us to **compute fidelity scores for all layers in a ResNet in approximately 5 minutes for ResNet-50 trained on Imagenet, and around 1 minute per layer to compute the optimal compensated mask $\delta_c^{\star}$.** Similarly, for Llama-2 models, **it takes approximately 30 minutes to compute Q matrices for all of the MLP layers, and approximately 2 minutes to compute $\delta_c^{\star}$.**
>
> **5. The experimental scope is somewhat limited:**
>
> We politely disagree that our experimental scope is limited. Commensurate with recent work [4], we have showcased the effectiveness of our method on two distinct tasks, on both vision and language models. Moreover, in Appendix C, we have provided thorough ablation studies supporting the validity of our hypothesis, highlighting that our key hypothesis holds across data splits and random seeds.
>
> **6. No exploration of the impact of k and η (key hyperparameters in HiFi selection).**
>
> We have already provided a discussion on the tradeoff between $k$ and $\eta$ in Appendix C.1.1. In particular, the Monte Carlo simulations presented in Figures 4-7 of the appendix illustrate the tradeoff between $k$ and $\eta$. Specifically, we plot $k$ vs $\eta$ for a variety of models, showing how different layers have smaller or larger subsets (of size $k$) that achieve high fidelity scores ($\eta$).
>
> **7. No discussion on the choice of B (pruning budget) for structured pruning or the sizes of forget set and retain set for classwise unlearning.**
>
> We point out that we have already provided a discussion on the impact of pruning budgets in Observation 1 (Section 3.2). Furthermore, in Appendix C,  we show, via Monte Carlo sampling, the impact of pruning on different layers in a variety of models. Note that the pruning budget $B$ can be thought of as the same as the size of the high-fidelity set $k$. For classwise unlearning, we only prune a small fraction of parameters, which varies from model to model.
>
> ---
>
> **Your valuable review has helped us strengthen our work, especially to show our strong empirical performance on SwinTransformers. We hope to consequently increase our scores, and are eager to discuss any further questions.**
>
>
> ## References
>
> [1] Torsten Hoefler, Dan Alistarh, Tal Ben-Nun, Nikoli Dryden, and Alexandra Peste. Sparsity in deep learning: Pruning and growth for efficient inference and training in neural networks. Journal of Machine Learning Research, 22(241):1–124, 2021.
>
> [2] Davis Blalock, Jose Javier Gonzalez Ortiz, Jonathan Frankle, and John Guttag. What is the state of neural network pruning? Proceedings of machine learning and systems, 2:129–146, 2020.
>
> [3] Jinghan Jia, Jiancheng Liu, Parikshit Ram, Yuguang Yao, Gaowen Liu, Yang Liu, Pranay Sharma, and Sijia Liu. Model sparsity can simplify machine unlearning. In Thirty-seventh Conference on Neural Information Processing Systems, 2023.
>
> [4] Chaitanya Murti and Chiranjib Bhattacharyya. DisCEdit: Model editing by identifying discriminative components. In The Thirty-eighth Annual Conference on Neural Information Processing Systems, 2024.

---

> > ### Comment · Reviewer_qfdv · 2025-08-05
> >
> > Thank you to the authors for providing the rebuttal. I have no further questions and will keep my current rating.

---

> ### Author Response · Authors · 2025-08-07
>
> Dear reviewer qfdv,
>
> As indicated by your previous response, we are happy that our rebuttal was able to address your concerns satisfactorily. In response to the concerns raised, we’ve made several changes that have strengthened the manuscript, especially,
>
> - Additional experimental evidence to strengthen our claims for Unlearning, specifically for transformer architectures.
> - Further specifying the setting of the problem, concerning the applicability of our proposed method on well-trained models.
>
> In light of the extension of the discussion period, and in the interest of further strengthening our work, we would be pleased to learn if there are any suggestions within the scope of the paper that would prompt the reviewer to increase their rating.

---

### Official Review · Reviewer_z8ND · 2025-07-07

**Clarity:** 2
**Significance:** 3
**Originality:** 3
**Rating:** 5
**Confidence:** 3

**Summary:**

This paper presents a new theoretical result bounding the global impact of removing a component from the neural network (in terms of how it affects the loss) with respect to its local impact (the change to the output of that specific component, such as a convolutional filter). This result depends on the neural network being Lipschitz continuous to start with. When that is the case, this result can be used to make it more manageable to perform structured pruning or classwise unlearning. In the first case, we would look for components with little effect if removed. In the second case, we would look for components that mostly affect performance in the class to be unlearned if removed.

**Questions:**

1) How do you characterize a well-trained model?

2) Where is the evidence claimed in Lines 49-51 that transformers are Lipschitz continuous?

3) There are many other approaches to structured pruning without access to the data or loss function that also measure the impact of components, such as [1-3]. Can you please contextualize your work with those and broaden your literature review accordingly?

[1] https://proceedings.neurips.cc/paper/2020/file/46a4378f835dc8040c8057beb6a2da52-Paper.pdf

[2] https://proceedings.neurips.cc/paper_files/paper/2021/file/e35d7a5768c4b85b4780384d55dc3620-Paper.pdf

[3] https://ieeexplore.ieee.org/document/9878423

4) How large can be the constants from Theorem 3.1? I am wondering if, either in theory or practice, these constants are sufficiently small to validate the approach.

5) There is a certain confusion about complexity classes in the paper: you talk about computing optimal subsets being NP-complete in Line 59, then say that this is a binary quadratic optimization problem in Line 168, mentioned to be NP-complete in Line 169, and then seem to refer again to this same problem being NP hard in Line 193. To be clear, optimization problems are typically NP hard. So my impression is that all other mentions are wrong. Are all of these mentions of the same problem? What is the complexity class of this problem?

6) Related to the item above, can you please rephrase footnote 2 in a clear and more detailed way?

7) What is the intuition for Theorem 3.7?

8) What do you mean by "the compensation term" in Line 228? Can you please rephrase that whole sentence for clarity?

9) When you talk about greedy selection in Lines 235-237, do you mean sorting the components from smallest to largest fidelity? That should be made explicit.

10) In Table 1 (which is actually two tables), what is the difference between the two rows for DFPC and Ours on ImageNet? Please make that clear in the paper.

11) For the same table as before, what exactly do you mean with columns CPU and GPU?

12) Across all tables, are the results in each row based on a single model or are they a summary of the results over multiple models with different random seeds?

13) I cannot understand Table 2. Can you please explain it?

14) Please consider these minor comments on the writing:
- Use "Lipschitz continuous" rather than "Lipschitz" alone, as that is not very formal and may confuse the reader if read out of context.
- Replace the word "tasks" with something like "purposes" in the abstract. Task may sound like the application of the neural network, such as classification for a specific dataset.
- Separate text from the reference citations with a space. If you do not want the citations in a separate line, use something like "x~\cite{y}".
- What is "F" in Line 259?

**Ethical Concerns:**

["NO or VERY MINOR ethics concerns only"]

**Final Justification:**

I cannot vouch for the correctness of the proofs, but the idea seems sound and I am satisfied with how the authors addressed the points raised.

**Limitations:**

There is a brief and reasonable acknowledgement about limitations at the end.

**Paper Formatting Concerns:**

None.

**Quality:**

3

**Strengths And Weaknesses:**

The authors make an interesting case by first presenting general theoretical results, refining them in a usable way, and then applying those results to practical problems. That makes the approach sound and reasonable.

With that said, I would have appreciated a little more intuition about the results, since I am not in a position to fully vouch for the claims and proofs. Moreover, some parts of the development are not entirely clear, and some of the experimental results are difficult to understand. I go in more detail about those issues in Questions.

---

> ### Author Rebuttal · Authors · 2025-07-31
>
> We thank the reviewer for their appraisal of our work and their insightful questions. We particularly appreciate their note on how we refine our theoretical results to apply them to problems of practical importance.
>
> We also want to take this opportunity to make an important clarification, that pruning and unlearning are consequences of our main finding: **the presence of a small number of statistically significant, high-fidelity components**. While we focus our experiments on the two applications listed, they are simply present to demonstrate that subset fidelity serves as a versatile tool for measuring component importance, and that it arises from the statistical properties of the model’s output alone, as opposed to the specifics of the application for which the importance is being computed.
>
> We now discuss the reviewer’s questions about our work, and hope that these address their concerns. If not, we are eager to further engage in a constructive discussion towards improving our work and scores.
>
> ## Questions
> **1. How do you characterize a well-trained model?**
>
> For our work, we make no formal characterizations of well-trainedness, and it suffices to treat well-trainedness to mean sufficiently high task performance, as is the case for all the models we experiment on. For our theoretical results, this allows us to leverage Lipschitz continuity in ResNets (for which it is established) and for pre-trained Transformers (for which we provide evidence, cf. our answer to Question 2 below). We will add a clarifying remark in the updated version of our manuscript to avoid confusion for the reader.
>
>
> **2. Where is the evidence claimed in Lines 49-51 that transformers are Lipschitz continuous?**
>
> Assumption B.2 (line 740) allows us to lower-bound the activations of normalization layers, which we then use to prove (in Lemma B.3, line 753) that said layers are Lipschitz continuous. We also provide empirical evidence in Figure 3 of Appendix C (on page 22) for the Lipschitz continuity of these layers across a selection of standard transformer-based models. We will add a reference to these experiments in the main body to enhance clarity.
>
> **3. There are many other approaches to structured pruning ... Can you please contextualize your work with those...**
>
> We appreciate the reviewer’s note of these relevant works, contextualizing against which has strengthened the position of our work, and we will add them to our Related Work section. For completeness, we provide said contextualization below.
> We emphasize that our method is not explicitly tailored for pruning, but presents a more fundamental approach to scoring components. However, in our applications to pruning, we differ from the cited works as follows:
>
> *Synflow* [1] targets **unstructured pruning at initialization**, and scores parameters to assess their “trainability”. Our work performs **structured pruning of well-trained** models and scores components according to their contribution towards the model’s output.
>
> *Leo++* [2], an exact compression method, can only be applied to **small ReLU-only feedforward neural networks** and is not generalizable to architectures like CNNs and Transformers, unlike our work. Additionally, they determine “prunability” alone instead of providing a relative scoring of components, with their algorithm being explicitly dependent on training data, while ours only requires distributional access.
>
> *CDG* [3] prunes CNNs while training, unlike our work, which is post-training pruning. Moreover, they score channels through gating the activations of a subset of channels within a layer, which requires access to training data. Our method does not require access to training data and is general enough to be immediately applicable to Transformers trained on language data.
>
> Importantly, **none of these works are directly applicable to unlearning.**
>
> **4. How large can be the constants from Theorem 3.1...**
>
> Theorem 3.1 only posits the existence of constants $C_l$. Empirically, we find that these constants are typically between 10-100 for ResNet50 trained on Imagenet. We will update the manuscript with exact details about these constants and how they are computed. In the proof of Theorem 3.1, we use an upper bound on the Lipschitz constants, which is very conservative. We emphasize that this is a conservative bound, and **reveals the non-trivial fact that even in the worst case, approximating global reconstruction error with the local one only incurs an error that is a linear function of the local reconstruction error.**
>
> **5, 6. There is a certain confusion about complexity classes in the paper: … Related to the item above, can you please rephrase footnote 2...**
>
> The complexities refer to two different problems: solving k-MFS and minimizing local reconstruction error.
> k-MFS (equation k-MFS above line 197), being an optimization problem, is indeed NP Hard. Minimizing local reconstruction error (below line 151) is exactly equivalent to the optimization variant of Max-Cut, whose decision problem (determining the existence of a cut of size k) is NP Complete. However, Max-Cut itself is NP Hard, and we will make this distinction clear in the manuscript.
>
> **7. What is the intuition for Theorem 3.7?**
>
> In our work, we make distributional access to data on which the model has been trained. Our work focuses on finding components of a model that are responsible for the model’s predictive power. We show that the problem of finding these components can be reduced to solving the k-MFS problem as stated in Definition 3.6. However, as noted in line 199, this is computationally infeasible. Theorem 3.7 makes the connection between the statistical properties of the data and the solvability of the k-MFS problem. In particular, in the scenario where the input representations of a layer are uncorrelated (a statistical property about the input representations of the layer), we can efficiently solve the k-MFS problem for that layer.
>
> **8. What do you mean by "the compensation term" in Line 228...**
>
> Thank you for pointing this out. We will update the manuscript with our clarification, which is as follows:
>
> ‘Compensation’ refers to the modification in the parameters by the application of the mask, aimed towards recovering performance. In our case, we perform linear compensation, scaling the parameters by the term specified in the second case of Equation FS, namely $1+((\mathbf{Q}^c_{C,C})^{-1})_{i}^{T} (\mathbf{Q}^{c} _{C, \bar{C}}) \mathbf{1}\_{n-k}$. In the case of pruning, this corresponds to scaling the retained weights within a layer according to the formula above. Compensation differs from recovery fine-tuning in that it is a closed-form update to the weights.
>
> **9. When you talk about greedy selection in Lines 235-237...**
>
> Thank you for bringing this up. Yes, by greedy selection, we mean the process of computing the set containing the **k-largest** singleton fidelity scores. This set represents the set of components that are either retained to perform pruning or discarded to perform unlearning. We will make this nomenclature explicit in Theorem 3.7 (on line 211, where the procedure is introduced) in our updated manuscript.
>
> **10, 11. In Table 1 (which is actually two tables), what is the difference ...? For the same table as before...**
>
> The two DFPC rows indicate the results after performing pruning for 30 and 54 iterations, respectively, as specified in their paper. The two “Our” rows correspond to the results when running ModHiFi-Prune with different sparsities, which result in different reductions in FLOPS and parameters in each row. The columns CPU and GPU refer to the relative speedups when running the model for inference on CPUs and GPUs, respectively. Specifications of the compute platform are presented in Appendix C.
>
> **12. Across all tables, are the results...**
>
> Following standard practices in this literature [4,5,6,7,8,9], the results are computed on a single trained model. We have provided ablation studies on the stability of subset fidelity to noise, data size, and over a variety of trained architectures, which can be found in Appendix C.
>
> **13. I cannot understand Table 2...**
>
> Table 2 lists the performance of Llama-2-7B (in the zero-shot regime) when pruned via the listed algorithms to the specified parameter sparsities. ModHiFi-P-WikiText and ModHiFi-P-Alpaca are two variants of ModHiFi-Prune that differ only in the dataset used for calibration, as described in lines 313-317. Performance is reported for WikiText perplexity (for which lower is better) and for accuracy on standard language modeling tasks [9] (for which higher is better).
>
> **14. Please consider these minor comments on the writing…**
>
> We thank the reviewer for pointing these out and will incorporate the suggested changes into the manuscript.
>
> **We thank the reviewer for their comments and their thorough review, as they have helped us strengthen our work. We hope to consequently increase our scores and are eager for any further discussion with them.**
>
> ---
>
> ## References
> [4] Shah et al. Decomposing and editing predictions by modeling model computation. ICML 2024
>
> [5] Jia et al. Model sparsity can simplify machine unlearning. NeurIPS 2023.
>
> [6] Narshana et al. Dfpc: Data flow driven pruning of coupled channels without data. ICLR 2023.
>
> [7] Ma et al. Llm-pruner: On the structural pruning of large language models. NeurIPS 2023.
>
> [8] Men et al. Shortgpt: Layers in large language models are more redundant than you expect. Preprint - arXiv:2403.03853, 2024.
>
> [9] Ashkboos et al.Slicegpt: Compress large language models by deleting rows and columns. ICLR 2024

---

> > ### Comment · Area_Chair_cc2X · 2025-08-04
> >
> > Dear Reviewer,
> >
> > Thanks for your comments. Please check if the authors' response answered your questions, e.g., if the response answered your concern about intuition of theorem and proofs.
> >
> > - AC

---

> > ### Comment · Reviewer_z8ND · 2025-08-04
> > **Follow up**
> >
> > I appreciate the well articulated responses from the authors.
> >
> > Regarding Tables 1 and 2, please update the text and tables to make it easier for the reader to understand what is going on.

---

> > > ### Author Response · Authors · 2025-08-04
> > > **Thank you for your response**
> > >
> > > Dear Reviewer z8ND,
> > >
> > > Thank you for your engagement with our rebuttal. We appreciate your comments, which will undoubtedly strengthen the final version of our manuscript. To clarify, conference policy prevents us from uploading a revised manuscript during the discussion period. However, we reaffirm our commitment to incorporating all the discussed changes, including the updates to Tables 1 and 2, into the camera-ready version.
> > >
> > > We would also like to encourage you to consider increasing our scores if your concerns have been addressed. We would be pleased to engage and clarify more concerns towards strengthening our work and scores.

---

> ### Author Response · Authors · 2025-08-07
>
> Dear reviewer z8ND,
>
> In light of the extended discussion period, we are eagerly awaiting your response. If you are satisfied with all the clarifications, we request that you consider reevaluating our work more positively. Your valuable comments have strengthened our work, especially
>
> - Helping make notation consistent throughout the manuscript
> - Additional experimental evaluation to assess the practical relevance of our theoretical bound. These experiments have revealed that our bounds are indeed practical. Indicating the constants in our bounds (Theorem 3.1) are not too severe and do reflect practical settings.
> - General improvement in the presentation of the work
>
> We would be pleased to learn if there are any suggestions within the scope of the paper that would prompt the reviewer to increase their rating.

---

> > ### Comment · Reviewer_z8ND · 2025-08-08
> > **Follow up**
> >
> > I cannot vouch for the correctness of the proofs, but the idea seems sound and I am indeed satisfied with how you addressed the points raised, so I am increasing my score (5) but with the same lower confidence on my own judgement (3).

---

> ### Author Response · Authors · 2025-08-08
> **Thanks to the Reviewer**
>
> We greatly thank the reviewer for raising their score and their more positive appraisal of our work. As we have mentioned previously, the reviewers' insightful suggestions have helped us improve the quality of our work. If there are any further clarifications, we would be happy to engage with you further.

---

### Author Response · Authors · 2025-08-09
**Summary Remarks to Reviewers and AC**

Dear AC and Reviewers,

We thank you for your participation in the discussion period. Your responses have helped improve the quality of our work in terms of novelty, robustness, and in comparison to prior work. As the discussion period comes to an end, we provide a summary of the additions and modifications to the paper that we will incorporate, per requests from each reviewer, to help discussions in the next phase of reviews:


### New Experiments

- Computing approximation error (Reviewer **z8ND**): We compute the constants in Theorem 3.1 and provide empirical evidence that the bound is practical for real-world settings, despite it being a worst-case one.
- Unlearning with fine-tuning (Reviewer **qfdv**): We present meaningful improvements over the state of the art, and in particular, achieve perfect unlearning with a superior remain-class accuracy when training for the same number of epochs as [1]


### Expanding the scope of relevant work

- Other structured pruning work (Reviewer **z8ND**): We compare our work against SynFlow, Leo++, and CDG, and find that all contextualize their work differently, either requiring explicit access to training data or existing in a setting where using training data isn’t meaningful (such as data-free Pruning at Initialization).

- Other reconstruction error-based work (Reviewer **Ze6v**): We find that the task-agnosticity of our analysis sets it apart from other work, allowing us to apply it to unlearning. Moreover, prior art does not identify groups of connected components within layers, while ours does.

- Comparing against [4] (Reviewer **17dD**): We find that [4]’s bound is more conservative than ours, and holds only for feedforward layers with Lipschitz activation functions, while ours extends to transformers. We also account for complex interconnections within networks.

- Estimating importances over the entire model (Reviewer **17dD**): We compare against [2], and note that their work requires both the loss function and access to training data, and isn’t architecture-agnostic. Moreover, we explicitly bound the propagation of local error across layers, making it a viable alternative to global error-based methods.

- Comparison against [3] (Reviewer **17dD**): [3] is an iterative scheme that computes pruning-specific scores, and its application to convolutional and other architectures is unclear.


### Presentation and Clarifications

Each review helped improve the clarity of our presentation. In particular, explicit nomenclature, better captioning, and clarity regarding our computational hardness assertions (Reviewer **z8ND**); presenting our algorithm’s complexity within the main body (Reviewer **Ze6v**); improved terminology and clarifying the role of the Monte-Carlo experiments (Reviewer **17Dd**).



**We are happy to note that all the reviewers appreciate the paper’s existing strengths. They have each noted the fact that every claim and assertion is backed by solid theory and thorough experimental validation. If there are any final concerns, we are eager to engage with the reviewers before the end of the discussion period.**

[1] Jia et al. Model sparsity can simplify machine unlearning. NeurIPS 2023

[2] Shah et al. Decomposing and editing predictions by modeling model computation. ICML 2024

[3] Shen et al. Numerical Pruning for Efficient Autoregressive Models, AAAI 2025

[4] NISP: Pruning Networks using Neuron Importance Score Propagation, CVPR 2018

---

### Note · Authors · 2025-08-13

Dear Reviewers and AC,

In this note, we summarize our key contributions and the discussion period, and request that it be read along with our previous responses. We thank you for taking time to prepare thoughtful reviews and engaging with us in the discussion period.

In prior art, specific model modification tasks, such as pruning and unlearning, have required computing importance scores for individual components, typically using the training data and loss function. However, in settings of practical interest, one does not have access to the training data or loss function used to train. Motivated by these constraints, we propose HiFi components, which we show are disproportionately responsible for model predictions. Crucially, these can be leveraged for both structured pruning and unlearning. We observe that in most layers, only a small number (~20%) of components are responsible for prediction. We propose an algorithm that can identify such HiFi components with only distributional access to data for a broad range of architectures (CNNs, ViTs, LLMs). Our results show that it finds HiFi components, using which we obtain state-of-the-art results on pruning and unlearning.

We summarize the notable additions and modifications borne from the discussions detailed in [our previous response](https://openreview.net/forum?id=lClK4uBxSG&noteId=WQaL1eNFeE
) below.

### Additional Experiments and Context

- Showed that our unlearning method outperforms the current state-of-the-art when training data and extensive fine-tuning are available.
- Computed the constants in Theorem 3.1, showing that the proposed bound is practical
- Added discussion comparing the merits of our paper with related works on model editing and structured pruning
- Highlighted the novelty of using reconstruction error beyond the application of structured pruning, namely for component attribution and classwise unlearning
- Discussed how Theorem 3.1 is less conservative than that presented in related work which doesn’t apply to models with complex interconnections or explicitly rely on the data distribution.

***

Each review helped improve our clarity of presentation, and all suggestions will be incorporated into the final manuscript. We thank the reviewers for their positive appraisals of, and satisfaction with, our work. We request that they take the strengths they identified and our discussions into account and evaluate our work positively in the next phase of discussions.

---

### Decision · Program_Chairs · 2025-09-17

**Decision:**

Accept (spotlight)

**Comment:**

The paper proposes a synthetic data–based method (named ModHiFi) for estimating feature importance in neural networks *without* access to training data. It introduces the HiFi metric for local reconstruction, applying it to structured pruning and classwise unlearning. Leveraging approximate Lipschitz continuity in pretrained Transformers, the method theoretically links reconstruction error to prediction degradation. Experiments on CNNs and Transformers (CIFAR-10, ImageNet) show ModHiFi achieves competitive performance without original data.

This is novel idea, solving a critical problem in large model training and fine-tuning. The authors have well addressed the questions and concerns by reviewers during discussion period. We decide to accept this paper. The authors are also required to appropriately polish the paper by incorporating the reviewers' comments (e.g., the additional experiments and context).